# Performance of the flood warning system in Germany in July 2021 – insights from affected residents

Annegret H. Thieken[1], Philip Bubeck[1], Anna Heidenreich[1], Jennifer von Keyserlingk[1], Lisa Dillenardt[1], Antje Otto[1]

[1]Institute of Environmental Science and Geography, University of Potsdam, Karl-Liebknecht-Strasse 24-25, 14476 Potsdam, Germany

*Correspondence to*: Annegret Thieken (thieken@uni-potsdam.de)

**Abstract.** In July 2021 intense rainfall caused devastating floods in Western Europe and 184 fatalities in the German federal states of North Rhine-Westphalia (NW) and Rhineland-Palatinate (RP) questioning their flood forecasting, warning and response system (FFWRS). Data from an online survey (n = 1315) reveal that 35% of the respondents from NW and 29% from RP did not receive any warning. Of those who were warned 85% did not expect a very severe flooding and 46% reported a lack of situational knowledge on protective behaviour. Regression analysis reveals that this knowledge is influenced by gender and flood experience, but also by the contents and the source of the warning message. The results are complemented by analyses of media reports and official warnings that show shortcomings in providing adequate recommendations to people at risk. Still, the share of people who did not report any emergency response is low and comparable to other flood events. However, the perceived effectiveness of the protective behaviour was low and mainly compromised by high water levels and the perceived level of surprise about the flood magnitude, while good situational knowledge and the number previously experienced floods helped performing more effective loss-reducing action. Dissemination of warnings, clearer communication of the expected flood magnitude and recommendations on adequate responses to a severe flood, particularly with regard to flash and pluvial floods, are seen as major entry points for improving the FFWRS in Germany.

## 1 Introduction

From 12 to 19 July 2021, Western and Central Europe witnessed widespread and intense rainfall caused by the low pressure system "Bernd" that led to severe flooding in Belgium, Germany, Luxembourg, and the Netherlands as well as further European countries in lower intensities (Schneider and Gebauer, 2021; Kron et al., 2022). In the western part of Germany, particularly in the federal states of North Rhine-Westphalia (NW) and Rhineland-Palatinate (RP), rainfall amounts totalled to more than 100 mm in 72 hours over large parts of these two most affected states with local maxima of more than 150 mm in 24 hours (Junghänel et al., 2021). This rainfall led to urban flooding in some bigger cities such as Cologne, Düsseldorf, and Hagen as well as to quickly rising flash floods in small and steep catchments in the middle hills, particularly around the Eifel mountain ranges (Dietze et al., 2022; Kron et al., 2022).

In all of Germany, 189 people lost their lives, thereof 135 in RP, 49 in NW, two in Bavaria, two in Saxony and one person in Baden-Wurttemberg. Even one year after the flood, two people were still reported missing. Severe damage of around € 33 billion occurred in the residential, commercial and industrial sectors as well as in the public sector and at infrastructures (Koks et al., 2022; Munich Re, 2022). Governmental disaster aid of an unprecedented amount of € 30 billion has been provided to support reconstruction and recovery in the affected areas. After floods in August 2002, June 2013 and May/June 2016, this

is the fourth flood over the past 20 years that caused damage of more than € 2 billion in Germany (Kron et al., 2022; see Table 1 for an overview on recent flood events in Germany). Even worse, the death toll in July 2021 by far exceeds the number of fatalities caused by former floods, which amounted to 21 in 2002, 14 in 2013 and eleven in 2016. A higher death toll caused by a water-related hazard was only recorded for a storm surge in February 1962 along the North Sea Coast with 347 fatalities in Germany, thereof 318 in the city of Hamburg. Consequently, failures in warning, alerting and evacuation processes have

been discussed already shortly after the event of July 2021 (Cornwall, 2021).

Internationally, the substantial reduction of global disaster-related fatalities per 100 000 people by 2030 is the first target of the Sendai Framework for Disaster Risk Reduction 2015-2030 (SFDRR). Since it is the primary goal of early warning systems to prevent fatalities during a disaster, the SFDRR also aims at increasing the number of countries with multi-hazard early warning systems in its seventh target (UN, 2015). Worldwide, the effectiveness of early warning systems to save lives was

impressively demonstrated in the flood-prone country of Bangladesh: while a cyclone in 1999 claimed around 10,000 deaths, warning and evacuation reduced the death toll to 38 lives in 2013 (Hallegatte et al., 2020). Recent cyclones confirmed the success of the warning and response system (Ferdous et al., 2020). For Europe, Hallegatte (2012) estimated that weather information and warnings have annually saved hundreds of lives and 460 million to € 2.7 billion Euros of losses, while creating even higher benefits by optimized production in weather-sensitive sectors.

Flood warning systems are more adequately termed flood forecasting, warning and response systems (FFWRS; Parker and Priest, 2012). As such, they include continuous monitoring and forecasting of precipitation and water levels, the detection of potentially hazardous situations, which should be linked to defined thresholds and rules on when, how and whom to warn in case of expected heavy precipitation or rising water levels including (pre-defined) statements that alert and inform civil protection and potentially affected people. For the overall success of a FFWRS, civil protection and affected parties have to

respond adequately and effectively to an unfolding flood situation (Parker et al., 1994; Parker and Priest, 2012), e.g., by erecting temporary water barriers, by evacuating people from heavily affected areas or by limiting access to inundated areas, e.g., by road closures. Warning is successful if all components function across spatial and departmental borders. In this process, creation and dissemination of warnings that trigger adequate and effective response is seen as major challenge (e.g., Cools et al., 2016; Kuller et al., 2021), in which various contents and formats of a warning message and different dissemination channels

can be distinguished (Kuller et al., 2021). Furthermore, trust among partners and in institutions plays a crucial role (Parker and Priest, 2012; Cools et al., 2016; Morss et al., 2016).

In Germany, flood warning systems have been established since the 1880s (DKKV, 2015). Currently, capacities and responsibilities for forecasting, warning and response are divided between the federal, state and local levels. At the federal

level, the meteorological service (Deutscher Wetterdienst – DWD) is in charge of weather forecasting and severe weather warnings, such as heavy precipitation. Flood forecasting and warning is, however, the task of the individual federal states and is organized differently as described by DKKV (2015) and Kreibich et al. (2017). After the severe flood of August 2002, the DWD introduced a fourth warning level to indicate very extreme weather events. Some federal states reorganized and centralized their forecasting and warning centres, e.g., Saxony, Lower-Saxony, and Thuringia (DKKV, 2015). In addition, data on flood water levels are displayed in a joint nationwide web-portal (www.hochwasserzentralen.de). These changes led to an improved warning situation during the river flood of June 2013 (Thieken et al., 2016; Kreibich et al., 2017).

To warn the general public is primarily the task of the local level, e.g., the district administrations. Since 2017, warnings can be disseminated via a Modular Warning System (Modulares Warnsystem – MoWaS) hosted by the Federal Office of Civil Protection and Disaster Assistance (Bundesamt für Bevölkerungsschutz – BBK) to a wide range of warning multipliers and dissemination channels like media operators and warning apps (e.g., NINA, KATWARN). Some districts and municipalities also use sirens or loudspeaker announcements to warn their population directly. The first nationwide alert day after the German reunification in September 2020 revealed how difficult it is to operate warning systems successfully. The federal Ministry of Interior declared the test a failure as the MoWaS messages and consequently also messages of warning apps were delayed due to technical reasons (BBK, 2020; Deutscher Bundestag, 2020). Subsequently, the system was improved and was tested successfully in NW in March 2021 (BBK, 2021). However, user data and views were not analysed (BBK, 2021).

Even if alerts function technically, there are many "potential deficiencies at each stage of FFWRS which transfer through their enchained processes" (Parker and Priest, 2012). Eventually, warnings can only avoid flood impacts – primarily fatalities, but also financial losses – if people in flood-prone areas as well as the local disaster management or civil protection receive and notice the warning in time, trust the warning, understand its contents, and know how to respond and behave adequately (Penning-Rowsell and Green, 2000; Párraga Niebla, 2015; Morss et al., 2016). Using German survey data from 2002 to 2013, Kreibich et al. (2021) showed the importance of residents' situational knowledge on protective behaviour ('knowing what to do') for flood damage reduction. Such situational knowledge is at least partly influenced by the warning message itself that should not only contain information on the hazard process, location, and time, but also some guidance on protective behaviour (Kuller et al., 2021). Therefore, an evaluation of a FFWRS should include how the population at risk perceived the warnings and whether they were able to respond adequately (Penning-Rowsell and Green, 2000). As part of a broader post-event investigation, this paper aims to analyse how the warning system in July 2021 performed – also in comparison to other flood events in Germany that are summarized in Table 1. The evaluation of the performance of the warning system is mainly based on an online-survey in the affected regions and focusses on three research questions (RQ): RQ1) How many people received a warning before they were in danger? RQ2) How well did people trust and understand the warnings? RQ3) How did people respond to the warnings and how did they perceive the effectiveness of their action?

As indicated by Thieken et al. (2022) for the river flood of June 2013 in comparison to the pluvial/flash floods of May/June 2016, the performance of Germany's FFWRS differs per flood type. For pluvial and flash floods in 2016, there was a higher a share of affected people who were not warned, warning times were shorter and the situational knowledge was poorly developed

among affected residents (Thieken et al., 2022). Given the severe impacts in 2021, we hypothesize that the performance of Germany's FFWRS in July 2021 was even worse than during recent pluvial and flash floods (see Table 1 for brief event descriptions) with regard to the dissemination of the warning messages and people's situational knowledge on protective behaviour. Since elderly people were considerably overrepresented among the flood fatalities of 2021 (Kron et al., 2022), we expect that the receipt of warnings, the situational knowledge and the perceived effectiveness of protective behaviour is influenced by the age of respondents next to the event's magnitude. The flood magnitude of July 2021 was exceptionally high as estimations of precipitation indices and of return periods of the discharge along the river Ahr revealed (Lengfeld et al., 2022; Vorogushyn et al., 2022). Therefore, we further hypothesize that damage-reducing behaviour was not perceived as effective by the respondents.

Following an explorative approach, we finally discuss as a fourth research question (RQ4) how to further improve the FFWRS based on the outcomes of the analyses and the views and wishes of the population affected in July 2021.

## 2 Data and Methods

Between 25 August and 17 October 2021, an online survey on the warning situation in July 2021 was conducted. The online-survey was advertised via Facebook, primarily in the two most affected federal states of North Rhine-Westphalia (NW) and Rhineland-Palatinate (RP), but the questionnaire was provided in SoSci Survey and hence accessible from outside of Facebook. In addition, a press release was sent to local newspapers in the area and all mayors were informed by e-mail about the survey with a plea to mention it in local newsletters. Overall, there was a response to all advertising activities. In total, 1348 people completely answered the survey, thereof 892 from NW and 423 from RP; Fig. 1 shows the districts with respondents from these two states. The remaining 33 cases could not be located due to missing geographic information, or were located in other federal states and thus omitted from further analyses. In this paper, first analyses of the data set are presented.

The socio-demographic characteristics of the subsamples are summarized in Table 2 and are compared to the general population per federal state as of 31 December 2020. With regard to gender, the subsample of NW is somewhat biased towards women (Chi-Square goodness of fit test, p = 0.0003), while the subsample of RP is slightly, but non-significantly biased towards men. With regard to age, the age group of 41 to 60 years is overrepresented in both subsamples and accounts for almost half of the respondents. Adolescents (15 to 20 years), who were not explicitly addressed by the adverts, and very old people (>80 years), who might not be reached by the online format, are clearly underrepresented in both subsamples (Table 2). However, both samples include respondents from all age classes and hence cover a wide range (NW: 15 to 88 years; RP: 20 to 83 years). Therefore, the sample is believed to provide answers to the research questions. However, conclusions with regard to gender or age have to be drawn with special care.

The questionnaire comprised 22 questions, of which several were taken from similar surveys that have been conducted after floods since 2002 (Thieken et al., 2017; Kreibich et al., 2017) allowing us to compare the data from 2021 to the recent past

and to explore whether warning in July 2021 was comparable to or worse than during other flood events. The events used for comparison and the available survey data are listed in Table 1.

In line with the research questions (RQ), the questionnaire used in 2021 addressed the following topics on RQ1: warning source (dissemination channel), information content, point in time when the first warning was received; on RQ2: assessment of the credibility of the warning on a six-point rating scale, the anticipated magnitude of the flood, the perceived knowledge on how to react adequately (situational knowledge on protective behaviour), as well as the perceived level of surprise by the magnitude of the event; on RQ3: types of immediate response actions and a perception of their loss-reducing effect on a six-point rating scale, since shortly after the event data on financial losses were not available. As potentially independent variables that might influence the performance of the warning process, it was asked how the water entered the building (flood pathway), the maximum water level at the building, the perceived impacts of the event on the neighbourhood and on the own household. In addition, two questions on previously experienced floods were posed. Furthermore, the postal code and the place of residence, the age and gender of the respondent as well as the size of their household was elicited as socio-demographic information. With regard to RQ4 people were asked to indicate on a six-point rating scale how much they appreciate currently discussed channels of warning dissemination and how important they regard different pieces of information to be contained in a warning message. At the very end, respondents could provide further information considered important as open answer. The full questionnaire is provided in the Appendix. As data post-processing, the corresponding federal state, as well as the official codes and names of the district and the municipality, were added to each case based on the reported postal code and place of residence. In addition, indicators on the warning source and the information content were calculated in accordance with Thieken et al. (2005). The warning source indicator captures through which channel/by whom respondents received a warning, ranging from 'no warning' and 'own search' to 'official warnings' from authorities or local disaster management. The warning information indicator reflects the reported pieces of information of the warning message. It ranges from 'no relevant information/no warning' to 'information on how to act and protect oneself'. All variable definitions, coding and summary statistics are provided in the Appendix (Table A1). Results on the warning process were verified by local media reports that were searched in a newspaper database and official warnings released by MoWaS in July 2021 as well as via the warning app KATWARN.

To identify entry points for improvements of the FFWRS we examined, whether we can identify factors predicting 1) the receipt of an official warning issued by authorities (or not), 2) the perceived situational knowledge on protective behaviour, and 3) the perceived effectiveness of performed emergency response using regression analyses.

In a first logistic regression analysis, we examined factors that potentially relate to the receipt of an official warning (yes/no). Official warnings include warnings from authorities or civil protection, calls to evacuate, messages from weather apps as well as sirens or sound trucks. As potentially explanatory factors, we included socio-demographic information (age, gender, household size and the federal state of the respondents), the number of previously experienced flood events (prior flood experience), the perceived impact of the 2021-event on the respondent's household, as well as different flood pathways, as reported by the respondents. As an intuitive interpretation of regression coefficients is difficult for logistic regressions, we

provide odds ratios as a measure of the effect size, which are easier to interpret. An odds ratio above 1 indicates that, as the explanatory variable increases, the odds (or likelihood) of the dependent variable occurring also increases. Conversely, an odds
ratio below 1 indicates that, as the explanatory variables increases, the likelihood of the dependent variable occurring decreases. A second linear regression model analysed factors that potentially relate to people's situational knowledge on protective behaviour. As explanatory factors, we entered information on the warning source (also referred to as 'channel', e.g., by Kuller et al., 2021) and the content of the warning messages, the perceived flood impact at the respondent's household (as proxy for the flood magnitude), the number of previously experienced floods, the perceived degree of being surprise by the flood
magnitude, as well as age, gender and the federal state as socio-demographic control variables. Although the quality of the warning source is considered to increase with every category of the warning source indicator (see Table A1), the different categories are still entered as dummy variables in this regression model. We report a linear model in section 3.2 because regression coefficients can be interpreted more intuitively and since results are largely similar in terms of significant predictors compared with the corresponding ordered logistic model that can be found in the Appendix (Table A2).

A third model tested how to predict the perceived effectiveness of performed loss-reducing action. The perceived damage-reducing effect was elicited by the following question: "In your opinion: How much could your response before/during the event and/or private precautionary measures reduce the damage?" Following that question, explanatory examples of risk-reducing behaviour were provided, like the use of flood-adapted material and the purchase of water pumps, to facilitate a consistent interpretation by the respondents. The question was again elicited on a six-point rating scale (1: "not at all" to 6:
"almost completely"; see also Table A1). In addition to the warning source and the warning information indicators, we added the perceived situational knowledge on protective behaviour and examined whether water depth experienced at the building, previous flood experience and perceived surprise of the flood magnitude related to the perceived effectiveness of risk-reducing behaviour. Additionally, age, gender and the two federal states were added as socio-economic controls. As proxy for the flood magnitude, we tested the perceived impact on respondent's household and the water level at the building. Since the water level
explained more variance, that model is presented in section 3.3. We again report a linear model in the text and provide the corresponding ordered logistic regression in the Appendix (Table A3).

In general, data from rating scales that are end labelled are usually assumed to be equidistant, i.e., interval-scaled, according to Porst (2014), which was recently confirmed by Höhne et al. (2021) for questions on income (in-)equalities. Therefore, the survey data were mainly treated as quantitative data although in principle the equidistance of each rating scale needs validation.

**3. Results and discussion**

**3.1 Receiving warnings**

As outlined in the introduction, a prerequisite of an effective FFWRS is that warnings officially issued by authorities reach the people at risk. In July 2021, 35% of the surveyed residents from North Rhine-Westphalia (NW, n = 892) and 29% of those from Rhineland-Palatinate (RP, n = 423) stated that they had not been warned. Fig. 2 puts these high numbers into the context

of former fluvial (left) and pluvial (right) floods in Germany. Since flood forecasting and warning is the responsibility of the federal states (see Introduction), data in Fig. 2 are distinguished per federal state and event year for fluvial floods, while for pluvial floods, for which severe weather warnings of the DWD are decisive, just the name of the most affected city and the year of the event are provided.

Since August 2002, Germany has experienced several fluvial floods, particularly in the southern and eastern parts of the
country (see Kienzler et al., 2015; Thieken et al., 2022; Table 1). Fig. 2 reveals that in 2002, 2005, 2010 and 2016 the share of the affected population that received no warning is in general comparable to the outcomes in 2021 with only small differences across different federal states, except for Saxony-Anhalt in 2002. The flood processes of these events are also comparable to the situation in 2021, i.e., they occurred mainly in the middle hills and partly showed a flashy character (Kienzler et al., 2015; Thieken et al., 2022; Table 1). In 2002, the flood then travelled further downstream and caused inundations along the river
Elbe, which had the character of a (huge) fluvial flood particularly in Saxony-Anhalt, where the warning situation hence improved (Fig. 2; Kreibich et al., 2017).

In contrast to the events in 2002, 2005, 2010, 2016 and 2021, the share of the population that was not warned in 2006, 2011 and 2013 dropped to around 5 to 10% in most of the affected federal states (Fig. 2), which can be regarded as a good performance of the FFWRS (Thieken et al., 2016). These latter floods can be primarily characterized as slowly rising fluvial
floods (Kienzler et al., 2015; Thieken et al., 2016; Table 1). Fig. 2 further reveals that during pluvial floods the warning situation is even worse: the share of the unwarned population amounts to more than 50%, but shows, however, some improvements over time (see also Rözer et al., 2016).

Altogether Fig. 2 suggests that the performance of the FFWRS in Germany greatly depends on the type of flooding and is particularly challenged by pluvial and flash floods. For most of the pluvial floods shown in Fig. 2 as well as for the rainfall
and subsequent (flash) floods in May and June 2016, lead times of just two hours were reported by Kind et al. (2019, p. 79) based on official warnings. Survey data from residents affected in 2016 resulted in a median lead time of just one hour (Thieken et al., 2022). In addition, the forecasted rainfall amounts underestimated the observed values by far (Kind et al., 2019, p. 79). These analyses illustrate the limits of rainfall forecasts for convective storms. In 2021, however, the flood-triggering low pressure system had been forecasted several days in advance, i.e., since Sunday, 11 July 2021, by the European Flood Alert
System (EFAS) as well as by the German weather forecasting system (DWD, 2021). Hence, the share of residents who received no warning should have been considerably lower than surveyed, although Saadi et al. (2022) illustrate the tendency of radar-based rainfall data from July 2021 to underestimate rainfall amounts and hence flood peaks.

Table 3 presents the results of the logistic regression explaining the receipt (yes or no) of an official warning as defined in section 2. As regression coefficients are difficult to interpret in logistic regressions, we provide odds ratios as effect sizes (see
section 2 for an explanation). In terms of socio-demographic characteristics, we find that men report higher levels of being officially warned than women (increased odds ratio of nearly 67%). No significant effect is shown for age, the household size and the federal state. Having experienced flooding prior to 2021, increases the odds of receiving a warning in 2021 by 23%, while perceived strong impacts of the flood on the household decreases the odds by 18%. In terms of flood pathways, we find

that fluvial flooding (marginally significant) and wildly flowing surface runoff increases the receipt of a warning (odds ratio

of 36% and 43%, respectively), while a dike or dam breach reduces the odds ratio of an official warning receipt by 36% (marginally significant). Respondents who observed no flooding in their immediate surrounding reported significantly higher levels of being officially warned. While the latter finding might sound counterintuitive at first, it may be explained by the fact that respondents who were not flooded themselves were not surprised by water intrusion and thus had more time to receive an official warning. In addition, they might not have been affected by power outages or break-downs of telecommunications

which were frequently reported in severely affected areas (e.g., by Koks et al., 2022). Overall, the explanatory power of the model is rather low with an explained variance in official warning receipt of 6.3%. Maybe general habits of media usage or a person's social network could further improve the model's explanatory power. The timing of the flood event might also have an influence, since people are harder to reach at night.

In many places affected in July 2021, flooding occurred in the evening of 14 July and during the night from Wednesday to

Thursday (15 July). 740 respondents (valid answers from NW: n = 474; RP: n = 266) provided the day on which they were warned for the first time (Fig. 3). In both federal states, most respondents, who were warned, did receive the first warning on Wednesday, 14 July 2021, (NW = 40% of valid answers; RP = 61%). The second most frequent day for receiving a warning was Monday, 12 July 2021 (NW = 23%; RP = 16%). Altogether, around 35% of the warned residents from RP had received their first warning before 14 July, while this share amounts to 50% in NW. By the end of 14 July 2021, the cumulative sums

rise to 95% in RP and 90% in NW (Fig. 3).

In fact, the heavily affected district of Euskirchen (NW) issued a first warning with expected rainfall amounts of 200 mm via MoWaS on 12 July 2021 (around 5 pm local time), which was updated twice on 14 July 2021. Most of the other districts issued a first warning via MoWaS in the course of 14 July 2021; this was accompanied by state-wide warnings for NW and RP. The severely affected district of Ahrweiler (RP) issued a flood warning in the early afternoon of 14 July 2021 via the app

KATWARN; at 7:35 pm a water level of more than 5 meter was forecasted for the river Ahr.

Due to missing independent data on the outreach of different dissemination channels, there is only anecdotical evidence to compare our survey data with. For example, in the most affected district of Ahrweiler (RP) around 18% of the residents have subscribed to the warning app KATWARN. In the survey around 20 % of respondents from this district reported warnings from this app. In addition, their reports on the time slot of the first warning matches well to the officially released warning

message between 2 and 3 pm (data not shown). So, the answers of the respondents in Fig. 3 are basically consistent with the release of official warnings and underline the need to improve timely warning dissemination. According to a media expert (pers. communication on 4 April 2022) considerably more people would have been reached if the warnings and the upcoming event had been addressed in the TV and radio programmes for several days by using easily interpretable stories and images. The fact that warnings of slow onset fluvial floods like the one in June 2013 weare much more successful (as shown by

Kreibich et al., 2017; Thieken et al. 2022 and in Fig. 2) was explained by the longer coverage in the media starting with stories of affected places and people in the upstream areas. In comparison to TV and radio coverage, coverage with mobile phones is

much higher. However, residents in Germany used to have to subscribe to warning apps such as KATWARN or NINA; a cell broadcast system was introduced in 2022.

**3.2 Trusting and understanding warnings**

An investigation of the performance of a FFWRS should involve an assessment of the credibility and comprehensibility of the warning message as these are crucial aspects for response (Morss et al., 2016; Párraga Niebla, 2015). In July 2021, the credibility of the warning was in general high, but also revealed some doubts: on a six-point rating scale (1: "the warning was totally incredible" to 6: "the warning was highly credible") 48% of the 841 respondents, who had been warned and answered this question, chose a 5 or 6 (NW: 47%, RP: 51%). Around 9% found the warnings incredible, i.e., chose a 1 or 2 (NW: 8%,

RP: 11%). This distribution is very different when it comes to the anticipated magnitude of the forecasted event – and thus the understanding people got of the upcoming event after having received a warning: on a six-point rating scale (1: "it will rain, but there's no problem" to 6: "torrential rain will cause widespread inundations, massive damage and life-threatening situations") just around 15% of the 856 respondents, who had been warned and answered this question, chose a 5 or 6 (NW: 15%, RP: 14%) and 29% (NW: 30%, RP: 26%) chose a 1 or 2. This underlines that the warnings failed to credibly communicate

the magnitude of the upcoming event. This is reflected by the perceived level of surprise about the flood magnitude: on a six-point rating scale (1: "the magnitude of the event didn't surprise me at all" to 6: "the magnitude of the event totally surprised me") just around 5% of the 877 respondents, who had been warned and answered this question, chose a 1 or 2 (NW: 5%, RP: 4%), while 86% (NW: 87%, RP: 84%) chose a 5 or 6. In many parts of the affected areas, the flood of July 2021 was larger than any flood that had been measured in the continuous discharge series (e.g. Apel et al., 2022; Saadi et al., 2022). Our data

underline that the flood magnitude was largely underestimated by the affected residents. In addition, some respondents complained that too many warnings on Covid-19 were disseminated via the most popular warning app NINA, which was tiring and lowered their attention to warning messages. Above all, in the week prior to the severe flood event there were already warnings for heavy rain in parts of the affected region, but no serious flooding happened. False alarms are known to commonly lower trust in warnings.

Warning can only avoid flood impacts – in terms of deaths, but also in terms of financial damage – if people know how to respond and how to behave adequately (Kreibich et al., 2021; Kuller et al., 2021). Thus, the situational knowledge about how to avoid dangerous situations or mitigate damage should be assessed to learn whether people achieved a deeper understanding of the warning and were able to translate the warning into action. In the survey, the perceived situational knowledge on protective behaviour was assessed on six-point rating scale (1: "Based on the warning, I didn't know at all how to protect

myself and my household from the flooding" to 6: "Based on the warning, I knew very well how to protect myself and my household from the flooding"). Fig. 4 shows the lack of this situational knowledge as assessed by respondents who reported that they had been warned before the flood hazard became relevant for them and chose a 1 or 2 on the rating scale mentioned above. Similar to Fig. 2, the answers of 2021 can be compared to former surveys and flood events. Again, severe and flashy floods like those in 2002, 2010 and 2016 perform the worst and are comparable to the values reported for the flood of 2021.

Some answers from the slow river floods of 2006 in Lower-Saxony and 2013 in Bavaria or Thuringia (see Fig. 4), suggest that the flood magnitude and/or the lack of experience might play a role, too. To identify more specific entry points for improvements, we hence analysed the influence of various factors on people's situational knowledge during a flood by means of a regression analysis as explained in section 2. The results are displayed in Table 4. The corresponding ordered logistic regression model, which considers the ordered nature of the dependent variable is provided in the Appendix (Table A2).

In terms of the warning source, results show that warnings issued by authorities have a significant positive influence on people's situational knowledge on protective behaviour, when compared with respondents that did not receive any warning (= base), which is in line with the literature review presented by Kuller et al. (2021). The other three warning source categories, i.e., own search, friends and neighbours, as well as nationwide or regional news, had no significant effect when compared to those without warning. A significant but rather weak positive effect is found for the warning information, i.e., if the warning

message contains information about adequate behaviour, people tend to perceive to be better informed and able to cope with the situation. A strong positive effect is observed for flood experience. As could be expected, people who had experienced one or more floods before the 2021-event report significantly higher levels of situational knowledge. Interestingly, this effect increases continuously with the reported number of previously experienced events (Table 4). In terms of the socio-demographic control variables, men tend to report higher levels of situational knowledge, while age had again no significant

effect. We also find that respondents from RP report higher levels of situational knowing than people from NW. Significant negative effects are found for the level of surprise and the perceived flood impact on the respondents' household, with surprise having the larger effect (Table 4). Apel et al. (2022) argue that forecasting the impacts, i.e., the potentially inundated areas, would have been helpful to communicate the extent of inundation and the life-threatening potential of the upcoming flood event. Overall, the model explains 33% of the variance in situational knowledge.

The findings were verified by a first content analysis of media reports on warnings before the event hit and of the official warnings that were disseminated via MoWaS or KATWARN. Some examples from the local press illustrate that even though warnings from the DWD were usually reported correctly, the corresponding advice on behaviour was, however, often too vague and seems – in hindsight – inappropriate given the high flood magnitude. Moreover, only around a third of media reports that mentioned warnings included recommendations on behaviour. For example, on 13 July 2021, the "Trierischer

Volksfreund" (region Trier, RP; Seydewitz, 2021) reported an extreme weather warning from DWD with up to 200 mm rainfall that may also lead to rising water levels in small rivers. The associated advice was that people living along small rivers and streams should monitor the situation and potentially undertake precautionary measures. However, what such measures involve was not specified. Another article published on 14 July 2021 (Ruhr Nachrichten, NW) similarly reported severe weather warnings for the district Unna (NW). The corresponding advice was to keep doors and windows closed and to store objects in

cellars on higher shelves. Finally, for the area of Koblenz (RP) the "Rhein-Lahn-Zeitung" (Lindner, 2021) reported on 14 July 2021 a warning of heavy rain and rising water levels that was associated with the advice for campers to be careful alongside rivers. More comprehensive advises on appropriate property-level measures were just found in the "Rhein-Zeitung" (RP) of 14 July 2021 and mentioned backflow preventers, water-proof doors and windows, as well as maintenance works.

Official warnings are usually accompanied by action recommendations. However, some recommendations seem not to fit to the situation that unfolded in July 2021. One example illustrating that the recommended protective actions were not adapted to the real situation is taken from the severely affected district of Ahrweiler (RP). Here, the app KATWARN warned against water levels of more than five meters at 14 July 2021 at 7:35 pm, which considerably exceeded the 100-year flood level of around 3.7 meter at the gauge Altenahr. However, the recommended protective actions for affected people were still to avoid cellars and underground car parks, not to drive on inundated streets and to clear drains and wells. These actions were clearly insufficient, since already at 8:30 pm houses in the municipality of Altenahr were reported to be half-way under water and flowing away at 10:40 pm (Frankfurter Allgemeine Zeitung, 15 September 2021; Staib and Steppat, 2021). Only at 11:09 pm the state of emergency was declared and people 50 m on both sides of the Ahr river were requested to leave their homes and evacuate by themselves – an advice which, at that time, was clearly too late and also dangerous. As one consequence, more than 330 people were rescued by helicopters from the roofs of their houses or from trees (Kron et al., 2022). In summary, the official warning messages seem to contain all necessary information, but were not adapted to the flood magnitude that occurred in July 2021.

### 3.3 Responses and perception of loss reduction

Whether a warning prevents or mitigates flood impacts, ultimately depends on performed damage-reducing actions. These are commonly divided into immediate emergency measures and (long-term) precautionary measures (e.g., Dillenardt et al., 2022). Since measures can be very diverse, we compared the percentage of people across different flood events who reported no (emergency) action or a continuation of their daily routines. Fig. 5 illustrates there is only a small percentage of less than 10% of flood-affected residents who dide not perform any emergency action during slow onset fluvial floods that occurred in 2006, 2011 and 2013 (Fig. 5). This share is a bit higher in areas that experienced flash floods, e.g., in 2002, 2005 and 2010, but not in July 2021 (Fig. 5). Higher percentages of inaction were only reported for pluvial flooding and the event of 2016. This might be due to the short lead times of just 2 hours (see section 3.1; Kind et al., 2019, p. 79) and might also be a reasonable life-saving behaviour given the rapid rise of water levels.

To gain insights into the perceived damage-reducing effect of risk-reducing behaviour, we ran a third regression model (Table 5). In line with Table 4, we again report a linear model for consistency and ease of interpretation and provide the ordered logistic model in the Appendix (Table A3). Results show that both the warning source and the warning information indicator did not relate significantly to the perceived damage reduction (Table 5). As it could be intuitively expected, higher situational knowledge on protective behaviour and floods experienced by the respondents prior to 2021 both relate to significantly higher levels of perceived damage reduction by the respondents. In contrast, respondents who experienced high water levels at their building and perceived the flood magnitude in 2021 as a surprise reported significantly lower levels of perceived damage reduction. In terms of the socio-economic control variables, respondents from RP reported significantly lower levels, which might be due to the very high flood magnitude. Overall, the model explains 23% of the variance. Altogether, the response of people affected in July 2021 is comparable to other (fluvial) floods, but seems to be compromised by the high flood magnitude.

### 3.4 Wishes for future warnings

In the online survey, respondents were finally asked about their views on warning contents and their wishes for (new) warning technologies (see section 2). Fig. 6 displays the mean assessments of the importance of different pieces of information by the respondents on a rating scale from 1 (not important at all) to 6 (very important) for both federal states. The data reveal that almost all information is regarded (very) important with slight compromises with regard to the timing and the expected amount of rainfall, comparisons with past events, potential impacts and information about detours, road closures or train cancellations. It should be noted that timing and height of water levels are considered more important than information on rainfall, which contrasts the media reports that focus more on severe weather warnings released by DWD than on hydrological forecasts. Moreover, Kuller et al. (2021) found in their literature review inconsistent results on the effectiveness of impact-based warnings (and the provision of uncertainties in warnings). Besides the contents shown in Fig. 6, Kuller et al. (2021) further recommend providing contact information. These were mentioned by respondents of our survey in the open answers.

In the future, affected residents are in favour of a countrywide installation of sirens and cell broadcast accompanied by enhanced media coverage (Fig. 7). There are only small differences between respondents from the two federal states. The lower values in RP for cell broadcast might be due to the fact that many people experienced power outages and a breakdown of telecommunication in July 2021 (Koks et al., 2022).

### 4. Conclusions and Recommendations

In Germany, the system of severe weather and flood warnings, better termed flood forecasting, warning and response system (FFWRS), has been improved over the past 20 years, particularly after the severe flood of August 2002. Although a good performance was achieved during fluvial floods in January 2011 and June 2013, our analyses show that the system is particularly challenged by pluvial and fast onset flash floods: around one third of residents at risk of pluvial or flash floods are not reached by severe weather or flood warnings. This was found across various federal states in Germany and across several fast onset floods including the event of July 2021. Hence, the FFWRS in Germany with responsibilities across multiple governance levels from the federal to the local level reacts in general too slowly to these events. To accelerate the dissemination of warning messages, widespread dissemination on mobile phones is an option; cell-broadcast has meanwhile been introduced in Germany and was tested in December 2022. Our online-survey of 2021 underlines, however, that residents from the regions affected in July 2021 tend to be in favour of sirens, probably since they do not depend so much on power and telecommunication networks than other dissemination channels.

The fact that the atmospheric system that triggered the floods of July 2021 was forecasted several days in advance, pinpoints to further weaknesses of the FFWRS. Warning messages with rainfall amounts are difficult to interpret. For a better understanding, rainfall needs to be translated into water levels and inundated areas. To make use of rainfall forecasts and to gain time for response, flood forecast models need to be improved and flood warnings need to be communicated. In Germany, flood forecasting is the responsibility of the federal states which have different models in place, while media often just refer

to weather warnings issued by the DWD. Hence, either a nationwide flood forecasting system should be set up or more investments in better regional or even local flood forecasting systems have to be made~~done~~. With regard to pluvial and flash floods, the need for implementing local warning systems, e.g., at small creeks, which have not been included in the flood forecasting system so far, has to be checked by local authorities.

To better highlight potentially affected areas warning messages should link flood forecasts to hazard maps or should directly provide estimated inundated areas for the forecasted event, particularly for severe events. In many areas affected in Germany, the flood of July 2021 was larger than any flood that had been captured by the continuous discharge series. The survey data underline that its magnitude was greatly underestimated by the affected residents; warning messages obviously failed to clearly communicate the flood magnitude and potential impacts. This aspect needs more in-depth investigations. For example, the reliable creation, dissemination, and understandability of maps that show the expected inundated areas should be tested, also against other pieces of information that were considered more important by respondents of our survey, e.g., affected places, timing of the flood peak or information on evacuations.

Our analyses show that it is important that official warnings, which usually include some action recommendation, reach the residents at risk since this generally improves their situational knowledge on protective behaviour. Still, our survey data reveal that up to 50% of the warned residents did not know what to do in July 2021. Again, similar percentages had been reported earlier for flash and pluvial floods. The results indicate that flood risk and crisis communication in Germany has focussed much on river, i.e., fluvial, flooding. Hence, efforts to communicate threats, mitigation options and adequate behaviour with regard to flash and pluvial floods have to be considerably enhanced. Examples from the local newspapers and official warning messages underline that warning messages have to be linked more consistently and regularly with recommendations on adequate behaviour and should better account for the anticipated magnitude of the unfolding flood event. For extreme scenarios such as the record-breaking flood of July 2021, more warning levels could be an option. Since each (official) warning level is associated with predefined recommendations on what affected parties should do to protect themselves (including translations to other languages), more warning levels could probably lead to a better communication of protective behaviour that is appropriate for the unfolding event. In general, the understandability of warning messages should be better tested and evaluated in future.

Our analyses show that previously experienced flooding facilitates all aspects along the warning and response chain, i.e., the receipt of a warning, the situational knowledge on protective behaviour as well as the (perceived) effectiveness of loss-reducing responses. Therefore, risk communication needs to better mimic flood experience and train successful behaviour. Since some analyses revealed gender-sensitivity, women should be addressed more specifically. Given the high death toll of 189 fatalities in Germany in July 2021, life threatening situations and their avoidance should be particularly communicated, although it is still unclear how many fatalities can be directly attributed to shortcomings of the FFWRS in July 2021. This should be a topic of future research to further improve the FFWRS and risk communication. Special attention should be given to elderly people due to their high percentage among the fatalities. However, in our analyses age was not a factor that influenced the receipt of

a warning, the situational knowledge on protective behaviour or the (perceived) effectiveness of responses. Due to the online-format of the survey this needs, however, further investigation.

The magnitude of the upcoming flood was probably underestimated by the responsible authorities, too. In some places, e.g.,
in the district of Ahrweiler (RP), this resulted in the fact that the state of emergency was declared too late and that evacuations of heavily affected settlement areas were initiated too late. In most German states, the declaration of the state of emergency is the responsibility of the district administrator since in most cases the district also has to bear the incurred costs. However, there is no mandatory training of district administrators in disaster management, who are elected politicians. Whether this is a primary weakness of the system needs some further research and thoughts. However, some federal states, e.g., Saxony, have
introduced a risk-averse decision strategy, meaning that there is an automatic declaration of the state of emergency if flood forecasts exceed the highest warning level. In other regions, local warning chains have been established so that a telephone chain is initiated from upstream to downstream along a river in case of flooding or another incident, e.g., pollution. The success and transferability of such approaches need further investigation. In general, a more continuous evaluation of the whole FFWRS would be an asset. Our analyses suggest that some shortcomings of the current FFWRS that were painfully revealed
by the severe event in July 2021 were not unique for this event, but generally apply to pluvial and flash floods. They could have been detected earlier by a better evaluation of the system after recent flood events including the perspective of the affected population. Altogether, future research should focus on how to design a FFWRS that alerts communities and residents at risk on time and clearly communicates flood magnitudes, threats and response options.

**Acknowledgements**

We thank all respondents that they participated in the survey despite the flood impacts they suffered from.

**Financial support**

The presented work was initially developed within the framework of the research project 'Urban resilience against extreme weather events—typologies and transfer of adaptation strategies in small metropolises and medium-sized cities' (ExTrass) funded by Germany's Federal Ministry of Education and Research (BMBF, FKZ 01LR1709A1) in collaboration with the
Research Training Group "Natural Hazards and Risks in a Changing World" (NatRiskChange) funded by the Deutsche Forschungsgemeinschaft (DFG; GRK 2043/2). Further analyses were performed in the framework of the project 'Governance and communication during the flood event of July 2021' (HoWas2021) funded by BMBF (FKZ 13N16230).

**Data availability**

The survey data are owned by the authors and can currently be provided upon reasonable request only.

**Author contributions**

AH, AT, AO, and PB developed the questionnaire and conducted the survey with the support of LD. AT, PB, AH, and LD processed and analysed the data. JvK searched and analysed media reports and official warning messages with support of AO. AT prepared the manuscript with contributions from all co-authors.

**Competing interests**

The contact author has declared that neither they nor their co-authors have any competing interests.

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

**Figures**

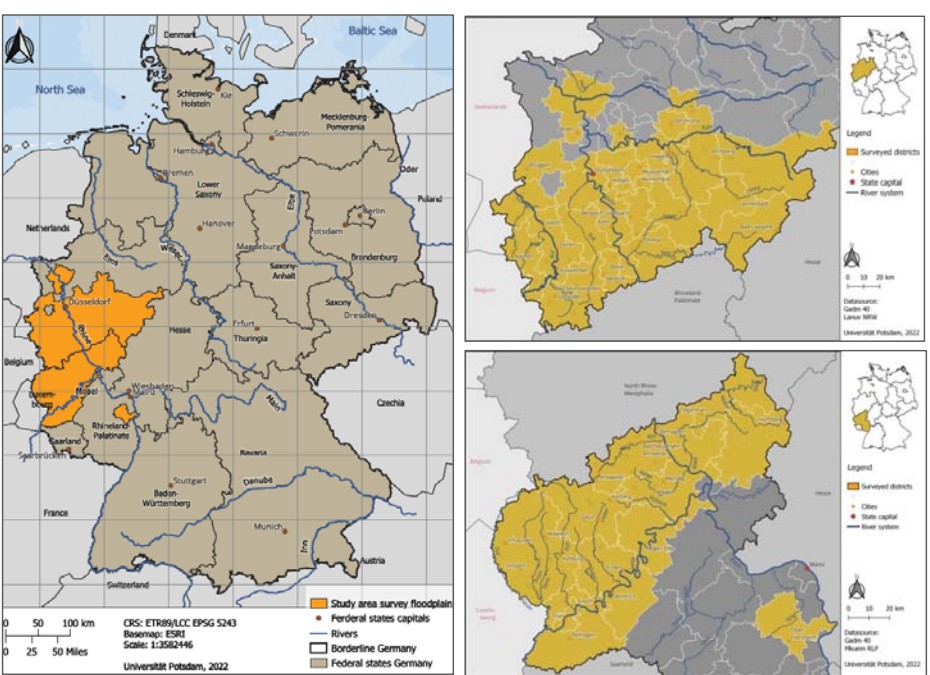

**Fig. 1: Overview map of Germany (left) highlighting the districts with respondents of the online survey in North Rhine-Westphalia (upper right) and Rhineland-Palatinate (lower right).**

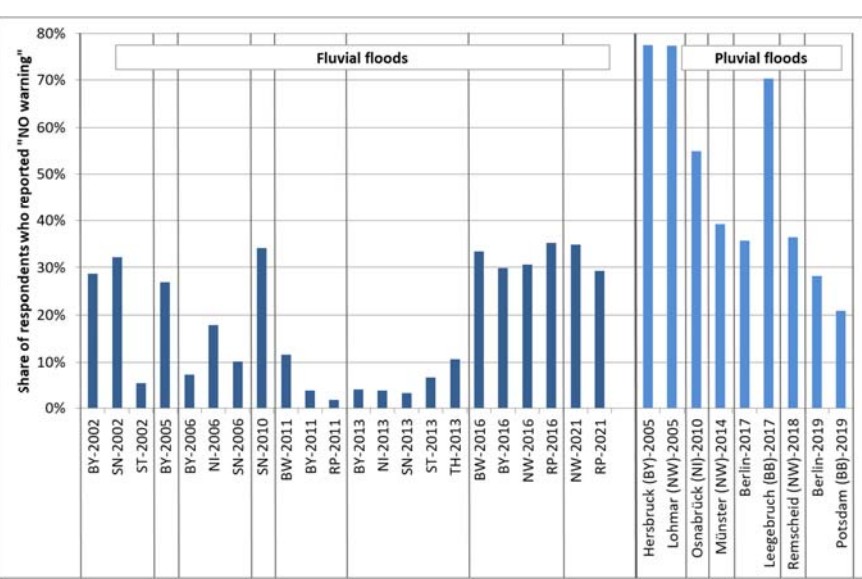

**Fig. 2: Share of respondents who reported that they had not been warned before the flood danger became imminent. Data are shown per flood event (year), federal state and flood type; left: fluvial floods from 2002 to 2021, right: some pluvial floods between 2005 and 2019 (abbreviations of the federal states: BB: Brandenburg; BW: Baden-Wurttemberg; BY: Bavaria; NI: Lower Saxony; NW: North Rhine-Westphalia; RP: Rhineland-Palatinate; SN: Saxony; ST: Saxony-Anhalt; TH: Thuringia; see Fig. 1 for geographic locations; see Table 1 for brief event descriptions).**


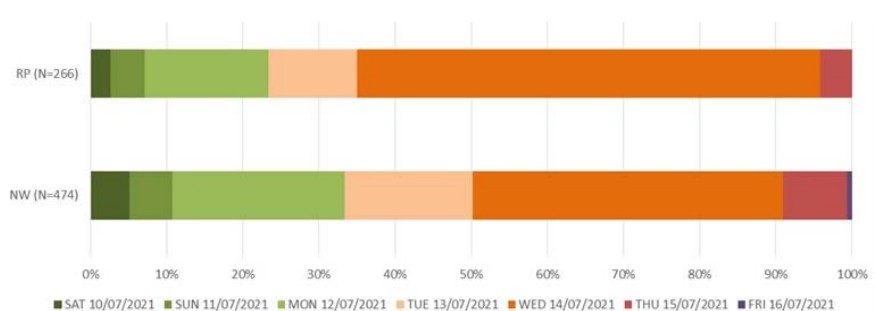

**Fig. 3: Day on which 740 respondents from North Rhine-Westphalia (NW) and Rhineland-Palatinate (RP) received a first warning.**

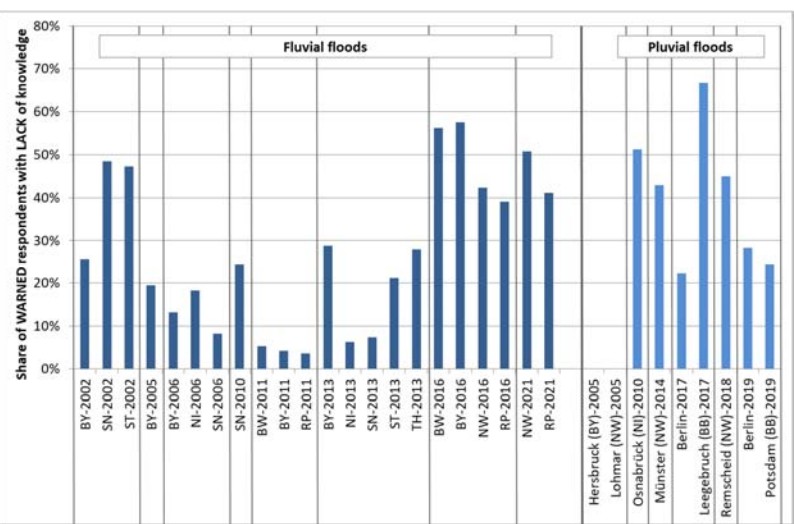


**Fig. 4: Share of respondents who were warned, but reported that they did not know (well) how to behave, i.e. how to protect themselves and their household against the flood. Data are shown per flood event (year), federal state and flood type; left: fluvial floods from 2002 to 2021, right: some pluvial floods between 2005 (no data) and 2019 (abbreviations of the federal states: BB: Brandenburg; BW: Baden-Wurttemberg; BY: Bavaria; NI: Lower Saxony; NW: North Rhine-Westphalia; RP: Rhineland-Palatinate; SN: Saxony; ST: Saxony-Anhalt; TH: Thuringia; see Fig. 1 for geographic locations; see Table 1 for brief event descriptions; note that in former surveys the scale was used in a reversed order; for this figure all data were aligned).**

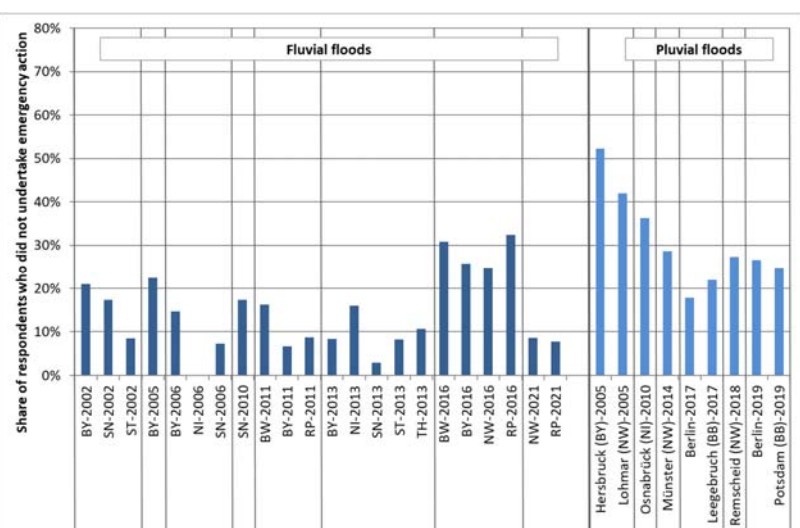

**Fig. 5: Share of respondents who reported no emergency action. Data are shown per flood event (year), federal state and flood type; left: fluvial floods from 2002 to 2021, right: some pluvial floods between 2005 (no data) and 2019 (abbreviations of the federal states: BB: Brandenburg; BW: Baden-Wurttemberg; BY: Bavaria; NI: Lower Saxony; NW: North Rhine-Westphalia; RP: Rhineland-Palatinate; SN: Saxony; ST: Saxony-Anhalt; TH: Thuringia; see Fig. 1 for geographic locations; see Table 1 for brief event descriptions; note that the question was phrased differently in surveys after the 2016-flood).**

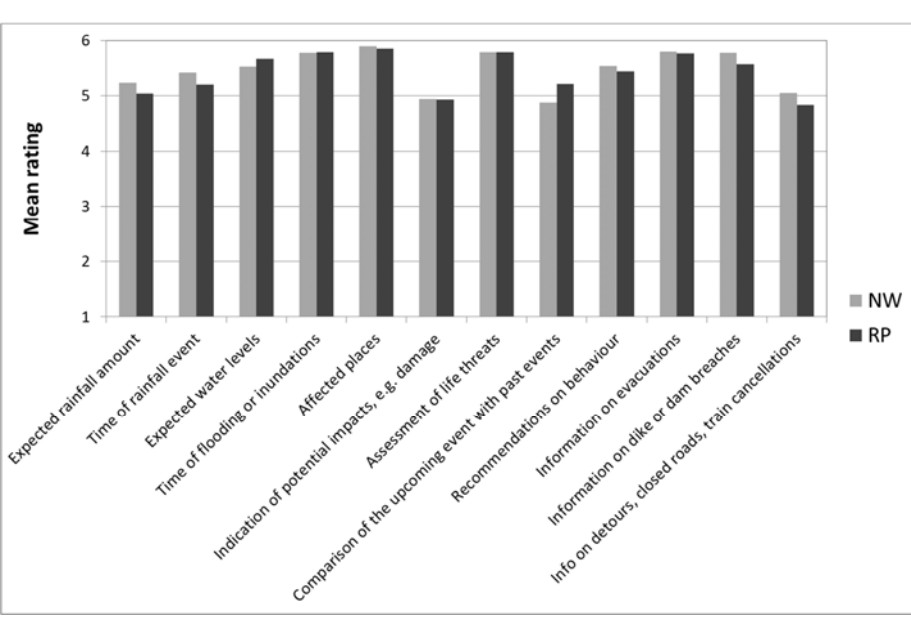


**Fig. 6: Mean rating of surveyed respondents with regard to the importance of different piece of warning information or content (NW: n = 837 to 882; RP: n = 404 to 418; rating scale from 1 'This piece of information is not important at all' to 6 'This piece of information is very important').**

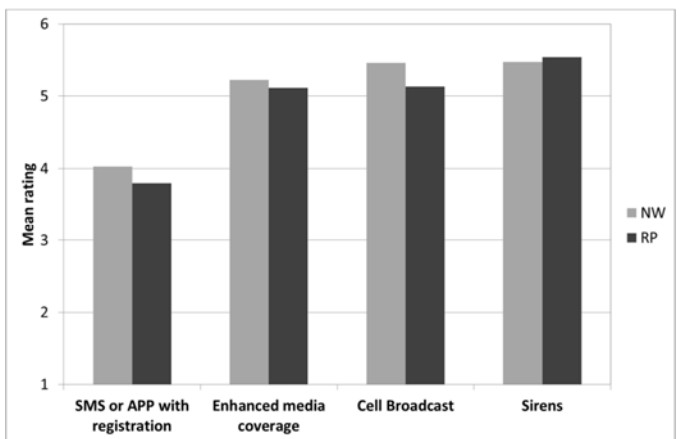

**Fig. 7: Mean rating of surveyed respondents with regard to future warning channels (NW: n = 837 to 882; RP: n = 404 to 418; rating scale from 1 'This measure is not helpful at all' to 6 'This measure is very helpful').**

**Tables**

**Table 1: Overview of recent damaging flood events in Germany and number of survey participants per federal state (compiled from Kienzler et al., 2015; Rözer et al., 2016; Spekkers et al., 2017; Thieken et al., 2016, 2017, 2022; Dillenardt et al., 2022; abbreviations of the federal states: BB: Brandenburg; BW: Baden-Wurttemberg; BY: Bavaria; NI: Lower Saxony; NW: North Rhine-Westphalia; RP: Rhineland-Palatinate; SN: Saxony; ST: Saxony-Anhalt; TH: Thuringia; see Fig. 1 for geographic locations).**

| Fluvial floods | Event description | Surveyed cases per federal state (if n > 24) | Field time of the survey |
|---|---|---|---|
| August 2002 | Flash floods in Bavarian and Saxon middle hills (e.g., in the Erzgebirge) caused by a Vb weather system with (very) extensive rainfall on saturated soils. In Germany, a record-breaking daily rainfall amount of 312 mm/24 hours was recorded. Flooding in the middle hills was followed by a slower onset flood along the river Elbe. | BY: 447 SN: 967 ST: 271 | 8 Apr to 10 Jun 2003 |
| August 2005 | Flash floods at the alpine foothills were caused by a Vb weather system with extensive rainfall that coincided with high preceding soil moisture and little snow cover in the Alps. | BY: 276 | 20 Nov to 21 Dec 2006 |
| April 2006 | Slowly rising river floods resulted from a high winterly snow cover that completely melted due to rapid temperature increase and was accompanied by heavy rainfall from westerly cyclones. | BY: 41 NI: 28 SN: 69 | 20 Nov to 21 Dec 2006 |
| August 2010 | Several flash flood waves were triggered by three consecutive fronts with heavy rainfall (due to a locked strongly meandering Jet Stream) and were intensified by a dam breach. Since measurements began in 1881, August 2010 was the wettest August in all of Germany. | SN: 305 | 16 Feb to 20 Mar 2012 |
| January 2011 | Slowly rising river floods in several catchments resulted from a high winterly snow cover, which melted due to a rapid temperature increase with heavy rainfall, followed by more intense rainfall. | BW: 43 BY: 75 RP: 57 | 16 Feb to 20 Mar 2012 |
| June 2013 | Local flooding was caused by a thunderstorm in May 2013 in Lower Saxony (NI). Widespread river floods were caused two weeks later by intense rainfall on highly saturated soils all over Germany. Record-breaking soil moisture was recorded in 40% of Germany by the end of May 2013. | BY: 239 NI: 50 SN: 523 ST: 593 TH: 216 | 18 Feb to 24 Mar 2014 |
| May/June 2016 | A series of (local) flash floods occurred between 26 May and 9 June 2016, when due to atmospheric blocking an extraordinarily high number of severe convective storms with low wind speeds leading to almost stationary and slow-moving cells and extreme local rainfall. The villages of Braunsbach (BW) and Simbach (BY) were particularly damaged. | BW: 195 BY: 191 NW: 85 RP: 71 | 28 March to 28 April 2017 |
| **Pluvial floods (urban flooding)** | | | |
| 29 June 2005 | Thunderstorms with heavy rainfall, storm gusts, lightning and hail developed along a boundary zone of colliding | Hersbruck (BY): 111 Lohmar (NW): 62 | 21 Nov to 19 Dec 2006 |

| | | | |
|---|---|---|---|
| | warm humid subtropical air from the southwest of Europe with cold and dry air masses from the north. | | |
| 26 August 2010 | A weather system (due to a locked and strongly meandering Jet Stream, see above) brought 128 mm of rain (i.e., 47% of the mean monthly precipitation of August) and overburdened drainage capacities in the city of Osnabrück (NI) leading to urban flooding. | Osnabrück (NI): 91 | 16 Feb to 20 Mar 2012 |
| 28 July 2014 | Extraordinary amounts of rain, i.e., 292 mm in 7 hours with a peak of 220 mm in <2 hours, were dumped on the cities of Münster and Greven (NW) due to an interaction of a stationary cold front with constantly incoming hot and humid air from the east resulting in widespread urban flooding. | Münster (NW): 510 | 20 Oct to 26 Nov 2015 |
| June/July 2017 | Local convective storms resulted in high rainfall amounts that overburdened drainage systems and caused inundations of urban areas. In the village of Leegebruch (BB) the water stayed for weeks due to its location in a low-lying area. | Berlin: 28 Leegebruch (BB): 91 | July 2019 to May 2020 (see Dillenardt et al., 2022) |
| Summer 2018 | | Remscheid (NW): 33 | |
| June 2019 | | Berlin: 64 Potsdam (BB): 105 | |


**Table 2: Socio-demographic characteristics of the sample in comparison with the general population per state as of 31 December 2020 according to Destatis (2021).**

| Gender | North Rhine-Westphalia (NW) | | | Rhineland-Palatinate (RP) | | |
|---|---|---|---|---|---|---|
| | number of respondents | % | % population as of 31 Dec. 2020 | number of respondents | % | % population as of 31 Dec. 2020 |
| male | 354 | 42.8% | 49.1% | 207 | 52.5% | 49.4% |
| female | 474 | 57.2% | 50.9% | 187 | 47.5% | 50.6% |
| **subtotal** | **828** | **100%** | **100%** | **394** | **100%** | **100%** |
| diverse | 1 | | | 1 | | |
| missing | 63 | | | 28 | | |
| **total** | **892** | | | **423** | | |
| age | n | % | % population without children | N | % | % population without children |
| 15-20 yrs | 10 | 1.1% | 6.9% | 1 | 0.3% | 6.6% |
| 21-40 yrs | 298 | 33.7% | 28.8% | 101 | 24.3% | 27.7% |
| 41-60 yrs | 435 | 49.2% | 33.0% | 235 | 56.6% | 33.1% |
| 61-80 yrs | 136 | 15.4% | 24.1% | 76 | 18.3% | 25.4% |
| >80 yrs | 5 | 0.6% | 7.1% | 2 | 0.5% | 7.2% |
| **subtotal** | **884** | **100%** | **100%** | **415** | **100%** | **100%** |
| missing | 8 | | | 8 | | |
| **total** | **892** | | | **423** | | |


**Table 3: Results of a logistic regression explaining the receipt of an official warning (n = 1115). All variable definitions, coding and summary statistics are provided in the Appendix, Table A1.**

| Explanatory Variable | Odds Ratio | Std. Err. | p | 95% conf. interval | |
|---|---|---|---|---|---|
| Age | 1.004 | 0.005 | 0.387 | 0.994 | 1.014 |
| Gender | 1.668 | 0.216 | 0.000 | 1.294 | 2.151 |
| Federal State | 1.075 | 0.077 | 0.310 | 0.934 | 1.238 |
| Perceived flood impact for household | 0.818 | 0.031 | 0.000 | 0.759 | 0.882 |
| Number of experienced floods prior to 2021 | 1.235 | 0.098 | 0.008 | 1.056 | 1.443 |
| Household size | 1.003 | 0.052 | 0.957 | 0.906 | 1.111 |
| No flood in immediate surrounding | 2.030 | 0.643 | 0.025 | 1.091 | 3.776 |
| Overloaded sewage water system | 0.816 | 0.122 | 0.175 | 0.608 | 1.095 |
| Wildly flowing surface runoff | 1.430 | 0.204 | 0.012 | 1.081 | 1.893 |
| Water ingress from toilets, floor drains etc. | 0.945 | 0.168 | 0.750 | 0.668 | 1.338 |
| Overflowing water body (e.g. river) | 1.361 | 0.239 | 0.079 | 0.965 | 1.919 |
| Dike or dam breach | 0.639 | 0.155 | 0.065 | 0.398 | 1.027 |
| Groundwater ingress | 1.127 | 0.168 | 0.425 | 0.840 | 1.510 |
| _cons | 0.192 | 0.151 | 0.035 | 0.042 | 0.893 |

Pseudo $R^2$ = 0.063

**Table 4: Results of the linear regression model predicting respondents' situational knowledge on protective behaviour (knowing what to do; ( n = 1097). All variable definitions, coding and summary statistics are provided in the Appendix, Table A1.**

| Explanatory Variable | Coefficient | Std. Error | p | 95% Conf. Interval | |
|---|---|---|---|---|---|
| Age | 0.003 | 0.003 | 0.285 | -0.003 | 0.009 |
| Gender | 0.378 | 0.082 | 0.000 | 0.216 | 0.540 |
| Federal State | | | | | |
| North Rhine-Westphalia | 0.000 | (base) | | | |
| Rhineland-Palatinate | 0.336 | 0.087 | 0.000 | 0.165 | 0.507 |
| Warning source indicator | | | | | |
| Not warned | 0.000 | (base) | | | |
| Own search | 0.113 | 0.291 | 0.697 | -0.458 | 0.684 |
| Friends or neighbours | 0.036 | 0.155 | 0.819 | -0.269 | 0.341 |
| National News | 0.273 | 0.217 | 0.208 | -0.152 | 0.699 |
| Official warning | 0.328 | 0.150 | 0.029 | 0.034 | 0.622 |
| Warning information indicator | 0.107 | 0.049 | 0.028 | 0.012 | 0.202 |
| Number of experienced floods prior to 2021 | | | | | |
| Never before | 0.000 | (base) | | | |
| Once | 0.525 | 0.125 | 0.000 | 0.279 | 0.771 |
| Twice | 0.702 | 0.195 | 0.000 | 0.320 | 1.085 |
| Three times | 1.466 | 0.324 | 0.000 | 0.830 | 2.101 |
| Four times or more | 1.510 | 0.309 | 0.000 | 0.903 | 2.118 |
| Perceived surprise | -0.491 | 0.045 | 0.000 | -0.580 | -0.402 |
| Perceived flood impact for household | -0.065 | 0.024 | 0.006 | -0.112 | -0.019 |
| _cons | 4.307 | 0.345 | 0.000 | 3.630 | 4.984 |

$R^2 = 0.33$

**Table 5: Results of the linear regression model predicting respondents' perceived damage reduction due to risk-reducing behaviour (n=1003). All variable definitions, coding and summary statistics are provided in the Appendix, Table A1.**

| Explanatory Variable | Coef. | Std. Error | p | 95% Conf. Interval | |
|---|---|---|---|---|---|
| Perceived situational knowledge (knowing what to do) | 0.205 | 0.034 | 0.000 | 0.139 | 0.27 |
| Warning source indicator | 0.004 | 0.04 | 0.915 | -0.074 | 0.082 |
| Warning information indicator | 0.055 | 0.051 | 0.287 | -0.046 | 0.155 |
| Age | -0.006 | 0.003 | 0.061 | -0.013 | 0.0 |
| Gender | 0.161 | 0.092 | 0.079 | -0.019 | 0.342 |
| Federal State | | | | | |
| North Rhine-Westphalia | 0.000 | (base) | . | . | . |
| Rhineland-Palatinate | -0.279 | 0.099 | 0.005 | -0.474 | -0.084 |
| Number of experienced floods prior to 2021 | 0.188 | 0.061 | 0.002 | 0.07 | 0.307 |
| Perceived surprise | -0.209 | 0.056 | 0.000 | -0.318 | -0.1 |
| Water depth | -0.213 | 0.026 | 0.000 | -0.264 | -0.162 |
| _cons | 3.626 | 0.434 | 0.000 | 2.774 | 4.478 |

$R^2 = 0.23$

**Questionnaire**

This short survey is aimed at **residents of the places affected by the heavy rain and flood event around July 14, 2021**.
Processing the survey should not take more than 10 minutes. Participation in the survey is of course anonymous. The results should help to clarify the warning situation in July 2021 and to improve the warning situation for future events. We therefore ask you to support us with your participation despite the current difficult situation.
Thank you very much!
___________

*Surveys can help to process what has been experienced, but can also lead to the event becoming a burden again. Please seek help in this case. If you need acute psychological help, please contact the BDP (Professional Association of German Psychologists) Flood Hotline: ☎ 0800 7772244.*
*Here you will find an overview of regional offers:*
*https://www.psychiatrie.de/flutkatastrophe-in-deutschland-seelische-unterstuetzung-fuer-betroffene-angehoerige-und-*
*helfende.html*

| 1. First of all, we would like to record the situation in your area: To what cause do you attribute the floods to in your immediate area in July? (Multiple choices possible) |
|---|
| ○    The sewage system could no longer drain the water on the road |
| ○    Overland water flow from streets or slopes |
| ○    Water overflow directly from the sewer system via drains, toilets and showers into the rooms below street level (e.g., into the cellar). |
| ○    Flooding caused by overflowing water bodies (i.e., nearby river or smaller body of water has overflown) |
| ○    Flooding as a result of a dike breach or dam breach |
| ○    Rising groundwater |
| ○    Other, namely: |

○    I do not know.

○    My immediate surroundings were not flooded. *(Go to question 3.)*

| 2. At the maximum water level: How high was the water approximately on the outside of the house? (This means the water level above the surface of the ground) |
|---|
| ○    There was no water in or around the house. |
| ○    There was only water in the cellar. |
| ○    Up to 0.5 meters |
| ○    >0.5 to 1 meters |
| ○    >1 to 2 meters |

○    >2 to 4 meters

○    More than 4 meters

---

3. Please think back to the hours before the event. How did you find out that the risk of flooding was becoming acute for you? (Multiple choices possible)

○    Severe weather or flood warnings by authorities or on-site disaster response (e.g., fire brigade, municipality, police)

○    Warning by evacuation call

○    Radio

○    Television (e.g., weather report or teletext)

○    Daily newspaper

○    Weather app

○    Severe weather app (e.g., Katwarn, NINA, Warnwetter App)

○    Siren or loudspeaker truck

○    Self-research on the Internet

○    Social networks on the Internet (e.g., Facebook, Twitter)

○    Through others, e.g., neighbors, acquaintances, colleagues, friends etc. (e.g., personal conversation, phone call, e-mail, WhatsApp)

○    Through my employer

○    Through care or educational institutions (e.g., school, daycare)

○    Other, namely:

---

○    I do not know.

○    I was not made aware of the danger at all / I was not warned.

---

4. Which of the following information did the warnings contain? (Multiple choices possible)

○    Time for the onset of heavy rain

○    Time for the occurrence of the high water or the flooding

○    Dangerous areas (place, district, etc.)

○    Expected amount of precipitation

○  Expected water level (e.g., height of the maximum water level)

○  Instructions and recommendations for self-protection (e.g., switch off the electricity, lock windows and doors, do not go into the cellar)

○  Information about evacuations

○  Information about dike or dam breaches

○  Assessment of the life-threatening nature of the situation

○  Information about diversions, road closures and / or train cancellations

○  Information on possible effects, e.g., damage

○  Comparison of the expected event with past events / floods

○  Other information, namely:

○  I do not know.

○  None of this information.


5. Approximately when did you receive the first warning? Please include the day and approximate time period that you were warned.

| **Day of the Warning** | | **Time** |
|---|---|---|
| ○ Saturday, 10 July | ○ | Before 9:00 a.m. in the morning |
| ○ Sunday, 11 July | ○ | 9:00 a.m. to 12:00 p.m. |
| ○ Monday, 12 July | ○ | 12:00 p.m. to 3:00 p.m. |
| ○ Tuesday, 13 July | ○ | 3:00 p.m. to 6:00 p.m. |
| ○ Wednesday,14 July | ○ | 6 p.m.to 9:00 p.m. |
| ○ Thursday, 15 July | ○ | After 9:00 p.m. |
| ○ Friday, 16 July | | |
| ○ I do not know. | ○ | I do not know. |

6. How credible did you think the warnings were?

Completely credible  ①  ②  ③  ④  ⑤  ⑥  Not at all credible

| 7. Based on the warnings, how did you assess the severity (magnitude) of the anticipated event? | | | | | | | |
|---|---|---|---|---|---|---|---|
| It will rain, but that is not a problem. | ① | ② | ③ | ④ | ⑤ | ⑥ | There is a storm with extensive flooding, damage and life-threatening situations. |

| 8. Did you know how you can protect yourself and your household from flooding before the risk of flooding became acute for you? | | | | | | | |
|---|---|---|---|---|---|---|---|
| It was completely unclear to me. | ① | ② | ③ | ④ | ⑤ | ⑥ | It was perfectly clear to me. |

9. When you became aware of the risk of flooding, what did you do? (Multiple choices possible)

○      I went about my daily activities without paying attention to the event.

○      I informed others (e.g., friends, acquaintances, family) or helped them.

○      I researched information about heavy rain and / or floods.

○      I took measures to reduce the damage (e.g., secure documents and valuables, put furniture up, erected water barriers).

○      I turned off electricity / gas in my house.

○      I went to a safe place.

○      I got help.

○      I prepared for an evacuation and packed up important documents and things.

○      Other, namely:

○      I do not know.

---

10. How badly was **your place of residence** affected by the heavy rain or flood event?

| Not affected at all | ① | ② | ③ | ④ | ⑤ | ⑥ | Very badly affected |
|---|---|---|---|---|---|---|---|


11. How badly was **your household** affected by the heavy rain or flood event?

| Not affected at all | ① | ② | ③ | ④ | ⑤ | ⑥ | Very badly affected |
|---|---|---|---|---|---|---|---|

---

12. In your opinion: How much were you able to reduce damage through your response to the event and/or private precautionary measures? *

| Not at all | ① | ② | ③ | ④ | ⑤ | ⑥ | Almost completely |
|---|---|---|---|---|---|---|---|
| | | | | | | ○ | I do not know. |

*Private precautionary measures include, for example, the use of flood-adapted building and construction materials, the installation of flood-proof heating, the purchase of pumps or water barriers, etc.*

---

13. How surprising did you find the magnitude of the event in your immediate vicinity?

| The magnitude of the event didn't surprise me at all. | ① | ② | ③ | ④ | ⑤ | ⑥ | The magnitude of the event totally surprised me. |
|---|---|---|---|---|---|---|---|

---

14. How often have you personally – before July 2021 – been damaged by floods?

| | | | |
|---|---|---|---|
| O Never before | O Three times | | |
| O Once | O Four times | | |
| O Twice | O More than four times | | |
| O Not specified | | | |

| 15. When was the last time you were affected by a flood (before July 2021)? |
|---|
| Year:                                    Month: |


| 16. To conclude, we would like to come back to the warning situation. Many options for adapting the warnings are currently being discussed. | | | | | | |
|---|---|---|---|---|---|---|
| How helpful do you think the following measures are? | Not helpful at all | | | | Very helpful | |
| Cell broadcast, i.e., automatic sending of a warning to all cell phones in a certain region without prior registration | ① | ② | ③ | ④ | ⑤ | ⑥ |
| Warning messages via SMS or APP with prior registration | ① | ② | ③ | ④ | ⑤ | ⑥ |
| Comprehensive installation of sirens | ① | ② | ③ | ④ | ⑤ | ⑥ |
| Increased reporting on severe weather and / or flood warnings and correct conduct in the media (radio, television) | ① | ② | ③ | ④ | ⑤ | ⑥ |

| 17. How important is it to you that the following information is included in severe weather warnings? | | | | | | | |
|---|---|---|---|---|---|---|---|
| | Not important | | | | Very important | | I do not know |
| Time for the onset of heavy rain | ① | ② | ③ | ④ | ⑤ | ⑥ | O |
| Time for the occurrence of the high water or the flooding | ① | ② | ③ | ④ | ⑤ | ⑥ | O |
| Dangerous areas (place, district, etc.) | ① | ② | ③ | ④ | ⑤ | ⑥ | O |
| Expected amount of precipitation | ① | ② | ③ | ④ | ⑤ | ⑥ | O |
| Expected water level (e.g., height of the maximum water level) | ① | ② | ③ | ④ | ⑤ | ⑥ | O |

| | | | | | | | |
|---|---|---|---|---|---|---|---|
| Instructions and recommendations for self-protection (e.g., switch off the electricity, lock windows and doors, do not go into the basement) | ① | ② | ③ | ④ | ⑤ | ⑥ | O |
| Information about evacuations | ① | ② | ③ | ④ | ⑤ | ⑥ | O |
| Information about dike or dam breaches | ① | ② | ③ | ④ | ⑤ | ⑥ | O |
| Assessment of the life-threatening nature of the situation | ① | ② | ③ | ④ | ⑤ | ⑥ | O |
| Information about diversions, road closures and / or train cancellations | ① | ② | ③ | ④ | ⑤ | ⑥ | O |
| Information on possible effects, e.g., damage | ① | ② | ③ | ④ | ⑤ | ⑥ | O |
| Comparison of the expected event with past events / floods | ① | ② | ③ | ④ | ⑤ | ⑥ | O |
| Other information, namely: | | | | | | | |

18. In order to be able to make statements about what the warning situation looked like for people in the different affected regions of Germany, it is important that we know where most of them live. Therefore, please enter your postcode and place of residence.

Postcode:                    Location:

19. How old are you?

______          Years old

20. Are you...?

O female            O male            O other            O not specified


21. How many people live in your household at all times, including yourself and all the children?

______          People

22. Do you have any further comments?

**Thank you for your participation!**

Thank you very much for taking your time for this survey. We wish you personally and the whole region a lot of strength for

the reconstruction. If you have any questions, please contact: extrass@uni-potsdam.de


**Table A1: Variable definition, coding and summary statistics of the data set containing all cases from North Rhine-Westphalia and Rhineland-Palatinate (n = 1315).**

| Variable | Definition | n | Summary statistics Mean (St. Dev.) OR percentages |
|---|---|---|---|
| Dependent variables | | | |
| Receipt of an official warning | Dummy variable indicating whether respondents received an official warning from authorities or local disaster response. | 1250 | Yes = 42.7% No = 57.3% |
| Situational knowledge on protective behaviour ("Knowing what to do") | Answer to the question: "Did you know how you can protect yourself and your household from flooding before the risk of flooding became acute for you?" 1= it was completely unclear to me to 6 = it was perfectly clear to me. Please note that the scale was reversed for Fig. 2. | 1302 | 2.62 (1.60) |
| Perceived effectiveness of risk reducing behaviour/measures | Answer to the question: "In your opinion: How much were you able to reduce damage through your response to the event and/or private precautionary measures?" 1= not at all to 6 = almost completely. | 1303 | 2.37 (1.58) |
| Independent variables | | | |
| Age | Age of the respondents in years | 1299 | 48.0 (13.2) |
| Gender | Gender of the respondent: 1 = female; 2 = male | 1224 | Female = 54.0% Male = 45.8% Non-binary = 0.2% |
| Federal State | Indication of the federal state of the respondent: 5 = North Rhine-Westphalia (NW); 7 = Rhineland-Palatinate (RP) | 1315 | NW = 67.8% RP = 32.2% |
| Flood pathway | Description of the flood pathway (multiple answers possible): no flood in immediate surroundings; overload of sewage water system; wildly flowing surface runoff; water ingress from toilets, floor drains etc.; fluvial flood, i.e. overflowing water body (e.g. river); dike/dam breach; groundwater ingress | 1315 | No flood = 6.6% sewage system = 46.8% Surface runoff = 43.0% Floor drains = 18.6% Fluvial flood = 76.3% Dike/dam breach = 9.2% Groundwater = 28.8% |
| Warning source indicator | Nominal index that indicates the source of the warning with 0 = no warning; 1 = own search; 2 = friends or neighbours; 3 = national news; 4 = warning issued by authorities. In case of several warning, the most credible source (0<1<2<3<4) was assigned. | 1250 | No warning = 34.8% Own search = 2.4% Friends = 14.7% National news = 5.4% Authority = 42.7% |
| Warning information indicator | Index that indicates the quality of the warning content with 0 = no warning/no relevant information; 1 = information on | 1246 | 0 = 40.9% 1 = 1.8% 2 = 43.7% |

| | | | |
|---|---|---|---|
| | detours, road blockages and/or train cancellation, evacuation; 2 = information on timing and intensity of rainfall, on (maximum) water levels, potential damage, and/or information on dike breaches; 4 = information on how to behave and protect oneself and/or information on the life-threatening situation. | | 4 = 13.6% |
| Number of experienced floods prior to 2021 | Answer to the question: How often have you personally - before July 2021 - been damaged by floods? 1 = never; 2 = once; 3 = twice; 4 = three times; 5 = four times or more | 1308 | 1.35 (0.87) |
| Perceived surprise | Answer to the question: How surprising did you find the magnitude of the event in your immediate vicinity? 1 = The intensity of the event didn't surprise me at all to 6 = The intensity of the event totally surprised me. | 1313 | 5.56 (0.97) |
| Perceived flood impact on own household | Answer to the question: How badly was your household affected by the heavy rain or flood event? 1 = not affected at all to 6 = very badly affected | 1313 | 3.50 (1.78) |
| Water depth | Answer to the question: "At the maximum water level: How high was the water approximately outside at the house?" 1 = There was no water in or at the building; 2 = There was only water in the cellar; 3 = up to 0.5 meter; 4 = more than 0.5 and up to 1 meter; 5 = more than 1 and up to 2 meter; 6 = more than 2 and up to 4 meter; 7 = more than 4 meter. | 1248 | 3.57 (1.82) |

**Table A2: Results of the ordered logistic regression model predicting respondents' situational knowledge on protective behaviour (n = 1097).**

| Explanatory Variable | Coef. | Std. Err. | p | 95% Conf. Interval | |
|---|---|---|---|---|---|
| Age | 0.003 | 0.004 | 0.530 | -0.006 | 0.011 |
| Gender | 0.493 | 0.115 | 0.000 | 0.268 | 0.717 |
| Federal State | | | | | |
| North Rhine-Westphalia | 0.000 | (base) | | | |
| Rhineland-Palatinate | 0.392 | 0.122 | 0.001 | 0.153 | 0.630 |
| Warning source indicator | | | | | |
| Not warned | 0.000 | (base) | | | |
| Own search | 0.124 | 0.429 | 0.773 | -0.718 | 0.965 |
| Friends or neighbours | 0.154 | 0.221 | 0.485 | -0.279 | 0.587 |
| National News | 0.527 | 0.294 | 0.074 | -0.050 | 1.104 |
| Official warning | 0.571 | 0.209 | 0.006 | 0.162 | 0.981 |
| Warning information indicator | 0.147 | 0.067 | 0.028 | 0.016 | 0.279 |
| Number of experienced floods prior to 2021 | | | | | |
| Never before | 0.000 | (base) | | | |
| Once | 0.680 | 0.173 | 0.000 | 0.342 | 1.018 |
| Twice | 0.901 | 0.271 | 0.001 | 0.370 | 1.432 |
| Three times | 2.010 | 0.448 | 0.000 | 1.132 | 2.888 |
| Four times or more | 2.001 | 0.442 | 0.000 | 1.135 | 2.867 |
| Perceived surprise | -0.648 | 0.068 | 0.000 | -0.780 | -0.516 |
| Perceived flood impact on household | -0.078 | 0.033 | 0.019 | -0.143 | -0.013 |

**Table A3: Results of the ordered logistic regression model predicting respondents' perceived damage reduction by risk-reducing behaviour (n = 1003)**

| Explanatory variables | Coef. | Std. Error | p | 95% Conf. Interval | |
|---|---|---|---|---|---|
| Situational knowledge ("Knowing what to do") | 0.296 | 0.046 | 0.000 | 0.207 | 0.385 |
| Warning source indicator | 0.029 | 0.052 | 0.574 | -0.073 | 0.132 |
| Warning information indicator | 0.07 | 0.067 | 0.293 | -0.06 | 0.2 |
| Age | -0.012 | 0.005 | 0.008 | -0.022 | -0.003 |
| Gender | 0.216 | 0.123 | 0.081 | -0.026 | 0.458 |
| Federal State | -0.224 | 0.068 | 0.001 | -0.357 | -0.092 |
| Number of experienced floods prior to 2021 | 0.212 | 0.081 | 0.008 | 0.054 | 0.37 |
| Perceived surprise | -0.248 | 0.073 | 0.001 | -0.392 | -0.104 |
| Water depth | -0.28 | 0.036 | 0.000 | -0.351 | -0.209 |