# Peer review of "Performance of the flood warning system in Germany in July 2021 – insights from affected residents"

_EGUsphere, 2022_

## Author Comment (AC1)

**Responses to comments on egusphere-2022-244**

**"Performance of the flood warning system in Germany in July 2021 – insights from affected residents" by Annegret H. Thieken et al., EGUsphere, 2022**

In the text below we provide a point-by-point response to the comments made by the referees. For clarity, our responses are written in Italics.

**Referee 1: Martijn Kuller (Referee)**

**https://doi.org/10.5194/egusphere-2022-244-RC1**

Referee comments

This paper presents the results of a survey held among the residents of a recent severe flood event in Germany. It is a valuable piece of work both for the academic community interested in risk communication and for authorities to draw lessons from and improve practices in the future. While the overall quality of the work sufficient to warrant publication, there are a few issues that need to be addressed. Most importantly, the structure of the paper could benefit from clearer research questions, hypotheses and a methods and results section following this structure.

*Response 1: Thank you for valuing our work. We will consider changes in the structure of the paper and will phrase our research questions clearer in the revised version as outlined below. Initially, we did not follow a distinct hypothesis-based research design, but were more explorative. However, since we had some implicit hypotheses, we will phrase them explicitly in the revised paper. We will also carefully re-consider the structure of the individual sections of the article.*

Overall the language is good and the paper reads well, but some phrases are rather "spoken" language than written language style (e.g., line 39: "save people's lives". I would rewrite to: "prevent fatalities" or "prevent loss of life". Second example in line 79: "…how the targeted population…". I would change to: "…warning perception by the target population…"). The article would benefit from review by a native English writer. Below I provide further details on this, as well as other comments by section, and then detailed comments in a table.

*Response 2: The revised manuscript will be proofread by a native speaker before resubmission. The sentences pointed out by Reviewer 1 will be rephrased accordingly.*

Introduction

This section has a clear structure and explains the background, the need for the research and the research aim clearly. However, detailed research questions and hypotheses are missing, which are needed to justify the methods used (types of data (indicators) gathered, methods of data collection etc.).

*Response 3: In the revised version we will explain our research approach and underlying hypotheses clearer. Accordingly, we will extend the last paragraph of the introduction. A draft reads as follows:*

*"Therefore, an evaluation of a FFWRS should include how the addressed population perceived the warnings and whether they were able to respond adequately (Penning-Rowsell and Green, 2000). As part of a broader post-event investigation, this paper aims to analyse how the*

*warning system in July 2021 performed – also in comparison to other flood events in Germany. The evaluation of the performance of the warning system mainly focusses on two aspects: 1) How many people did receive a warning before they were in danger? 2) How well did affected people understand the warning and behaved accordingly? 3) How did the FFWRS perform in comparison to other recent pluvial and fluvial floods in Germany?*

*As highlighted by Thieken et al. (2022) for the river flood of June 2013 in comparison to the pluvial/flash floods of May/June 2016, the performance of Germany's FFWRS depends on the flood type. For pluvial and flash floods in 2016, there was a higher share of affected people who was not warned, warning times were shorter and the factual knowledge on adequate behaviour was poorly developed among affected residents. Given the severe impacts in 2021, we hypothesize that the performance of Germany's FFWRS in July 2021 was even worse than during recent pluvial and flash floods.*

*Since elderly people were considerably overrepresented among the flood fatalities of 2021, we expect that age is an important explanatory variable for the performance of the warning system next to the severity of the event.*

*Following an explorative approach, we also discuss how to further improve the FFWRS based on the outcomes of the analyses and the views and wishes of the population affected in July 2021 as mentioned in the survey."*

*The results section is already structured with regard to these questions/hypotheses. The methods section will be adapted.*

Data and Methods

It is stated that the survey was distributed via facebook, and a press release was sent through the newspaper. However, it seems like the survey could only be accessed through facebook. Why did the authors choose this method, and how did they make sure this didn't lead to a biased sample, as facebook is only used by a specific demographic, leaving out other specific demographics that might be important. Beyond age, general techsavvyness and online presence play a role in the accessibility of the survey to people using facebook. If the authors somehow tried to correct for this bias, please provide some information about how this was done. If not, please discuss how the method might have influenced the outcome and what this means for the conclusions.

*Response 4: This is a misunderstanding. The online questionnaire was accessible through a link outside of Facebook. Facebook was just used to advertise our survey. To avoid bias we also used traditional media like newspapers to draw attention to our survey and directly informed all mayors in the affected regions about our survey together with a plea to mention the survey in local newsletters. We saw a response to all advertising activities in the number of respondents to our survey. The age range presented in the paper illustrates that we managed to address elderly people, too, who are, however, underrepresented in the sample, although elderly people are increasingly online and use internet sources*

*We would like to highlight that we decided for an online survey in order to allow respondents a voluntary participation in the survey. This was important to us given the rather short time-span after the flood event, when many people were still in the immediate recovery process.*

Also, it is explained what the questions in the survey were regarding, however I am missing some explanation as to why these questions were posed (what were the exact research questions the authors want to answer with the set of questions they selected, including the

demographical questions), and what were they expecting to find? Part of this information could already be provided in the introduction. Furthermore, is the survey accessible to the reader somehow? It would be good to add them to supplementary materials for reference and future use by the readers.

*Response 5: With the revised introduction the content of the questionnaire will be more understandable. We will better explain the links between research questions and survey questions in the revised version. A translated version of the questionnaire can be provided as supplement if wished.*

The Likert scales used in the survey are analysed statistically, using the answers as numerical values. Although this is common practice, it is strictly speaking not valid, as these numbers are only category indicators, and not actual numerical values of the answer, and can thus not be used for statistical analysis (drawing mean, sd etc).

*Response 6: We use scales that are only verbalized at the end points as shown in Table A1. According to the (German) textbook of Porst (2014: Der Fragebogen, 4th edition, chapter 6), data from such a scale can be regarded as interval-scaled data (being ordered with equal distances between the values, but without having an absolute zero point). Therefore, it is valid to treat them as numerical data as it is also common practice, as also stated by the reviewer.*

Results and discussion

If the factors investigated and presented in Table 2 explain so little in the receipt of an official warning, what then is (or could be) the most important predictor for this? All the results are self-reported data from the survey. I wonder how this self-reported data compares to behaviour and if this could be discussed in this section (e.g., how do the numbers of "knowing what to do" compare to the observed behaviour of people on the day?). Would there be any such comparison possible here in the discussion?

*Response 7: Just for clarity: Table 2 shows the results of the regression on the receipt of the warning, while Table 3 shows the regression on factual knowledge (knowing what to do). For empirical data in social sciences, the latter has a good R² of 33 %.*

*We agree that it would be great to have some independent data to compare our results with. Unfortunately, there is only anecdotal evidence. For example, in the most affected district of Ahrweiler in Rhineland-Palatinate around 18 % of the population have subscribed to KATWARN, the commonly used warning app in this district. A more detailed analysis of our data for this district reveals that around 20 % of the people from this district in our survey report that they had been warned by this app. In addition, the time of having received the first warning as reported in the survey also matches to the officially released warning message.*

*Furthermore, during an interview with a representative of the regional broadcast we discussed the percentage of people, who were not warned. That number was then compared to data on the general coverage of TV and radio. According to the media expert a higher percentage of people would only have been reached if the warnings and the upcoming event had been addressed in the programme for a few days before the event happened and by using easily interpretable stories and images. The fact that warnings of slow onset fluvial floods like the one in 2013 are much more successful (as is shown in Thieken et al. 2022 and in this paper, too) was explained by the better coverage in the media for several days and first stories of affected places and people in the upstream areas.*

*Hard facts and data are, however, absent, but the aspects mentioned above can be briefly mentioned in the revised paper. In general, behavioural observations during the actual event would have been neither possible nor ethical.*

The results presented in Figure 5 show very little variation. This is a common problem with asking respondents what they want, choosing from a list of options: they are going to want it all. Such results are not very helpful when resources are scarce and trade-offs have to be made (in this case, we want to keep a warning clear and concise). A better way to measure such preferences could therefore be asking respondents to rank or to distribute a limited number of points among the options.

*Response 8: Thank you for this remark and the suggestion which we will consider in a next survey on this topic. Still, we think that our data are of use, e.g., with regard to the expected amount of rainfall. Here a clear mismatch between the frequency of this information in the received warning messages and the wishes for the future becomes apparent.*

A section on further research is missing. I already made some recommendations in this review, e.g., comparing self-reported information to actual behaviour (if this is not going to be implemented by authors).

*Response 9: We will add a dedicated paragraph with ideas on future research in the concluding section.*

Conclusions

Because the research questions weren't completely clear (see other comments), the conclusion doesn't naturally follow from the rest of the paper. When this issue is resolved, the conclusion might be written in a more concise manner, going back to these research questions.

There are many references to the main text as well as other papers in the conclusion. I recommend rewriting the conclusion according to my above instructions, and avoid any references, as a conclusion should stand on its own, not repeat the results, but rather state the significance and meaning of results. As it stands, the conclusion mostly summarises the main text.

*Response 10: Thank you for this suggestion. However, we partly disagree. A conclusion should or could start with repeating the research aim and the main findings, then proceed with implications (while accounting for the inherent uncertainties) and finally addressing future research topics. Following this structure, references to other works or to tables/figures in the paper might help the reader to connect things. According to the reviewer's preferences we will, however, reduce the number of references in the concluding section.*

Furthermore, I think the paper could benefit greatly from a separate section (within conclusion or at the end of the discussion) that summarises the most important recommendations from this research. This research is very practical and applied, and has great potential to aid authorities around the world.

*Response 11: In our view these recommendations are part of the conclusions as outlined above. We will pay attention to better address these in the revised version.*

Please find detailed comments in the table below:

Line 39-40: One or two sentences discussing why the performance might have been so good for these events could benefit this section here.

*Response 12: We will add some figures on the performance of FFWRS in the revised version.*

Line 153-158: This belongs more in the methods section, see previous comments.

*Response 13: These sentences will be shifted to the method section.*

Line 159: Results are presented for which no methods have been described (logistic regression is mentioned). This should be added to the methods section (also see previous comments about methods).

*Response 14: Logistic regression is a standard method. Therefore, we didn't mention it explicitly in the method section, but we will briefly do so in the revised version together with the paragraph from lines 153-158.*

Line 160: Please explain how to interpret odds ratios.

*Response 15: We apply an ordered logistic regression. The interpretation of the ordered logit coefficient is that for a one unit increase in the predictor, the response variable level is expected to change by its respective regression coefficient in the ordered log-odds scale, while the other variables in the model are held constant. As an intuitive interpretation is difficult due to the log-scale, we consider to provide odds ratios in the revised version of the manuscript as a measure of the effect size, which are easier to interpret. An odds ratio above 1 then indicates that, as the explanatory variable increases, the odds (or likelihood) of the dependent variable occurring also increases. Conversely, an odds ratio below 1 indicates that, as the explanatory variables increases, the likelihood of the dependent variable occurring decreases.*

Table 2 and 3: From the table it isn't clear how gender impacts the prediction on knowing what to do. Is it male or female that increases the knowing what to do (explain in the table).

*Response 16: The coding is explained in Table A1, but we will repeat the most important information in the headings of Tables 2 and 3.*

Line 184: Is there data available to make a more precise comparison between the warnings issued by the different authorities and the warnings received by the population as reported through the survey? This could shed light on the effectiveness of the dissemination by the authorities. (a hint to this information is given in the text in lines 183-184).

*Response 17: Please have a look at response 7 above. We haven't done this regional analysis for the whole region, which is quite large, but we could highlight one example to illustrate the finding.*

Line 189: The authors state that they find the trust in the credibility high. I would argue that 9% of people not believing a warning is quite a high number in an emergency situation, and less

than half of the people stating they find the warning highly credible is to my perception quite low. From a government-issued warning, I would strive for credibility numbers close to 100%.

*Response 18: We agree and will rephrase the sentence. There's anecdotal evidence that a lot of people distrusted the predicted severity, not the warning itself.*

Line 198: This sentence is confusing, please re-write.

*Response 19: The sentence now reads: "In addition, some respondents complained that too many warnings on Covid-19 were disseminated via the most popular warning app NINA, which was tiring and lowered attention to warning messages in general. Above all, in the week prior to the severe flood event there were already warnings for heavy rain in parts of the affected region, but had no serious (flood) impacts happened.*

Line 210: This is more a concluding remark.

*Response 20: This aspect will be included in the concluding section.*

Fig. 5 and 6: Meaning of the numerical scale points need to be explained in the caption. Furthermore, as indicated before, it is strictly speaking not correct analysing and presenting this data in this way.

*Response 21: Our apologies for this. The meanings of the scales are explained in Table A1, but will be repeated here. See our comment (response 6) on the validity above.*

Line 298: It is not common to refer to tables and figures in the conclusion if not strictly necessary.

*Response 22: see response 10*

*Thank you for the detailed comments to our paper.*

---

## Author Comment (AC2)

**Responses to comments on egusphere-2022-244**

**"Performance of the flood warning system in Germany in July 2021 – insights from affected residents" by Annegret H. Thieken et al., EGUsphere, 2022**

In the text below we provide a point-by-point response to the comments made by the referees. For clarity, our responses are written in Italics.

**Referee: Anonymous Referee #2**

**https://doi.org/10.5194/egusphere-2022-244-RC2**

Referee comments

This paper provides very important insights into the early warning, preparedness, and response of the German flood event in 2020. The major aim of the paper is to analyze the operation of the warning system in July 2021 and to compare survey results to survey results of historic floodings in Germany. The results presented in this paper will be of high value to improve the disaster management in Germany.

I agree with the major recommendations provided in the comment RC1; thus, below, I will provide only a few suggestions in addition to the already stated recommendations.

*Response 1: Thank you for valuing our work.*

Introduction/Methods

The introduction provides a clear and in-depth overview on the event, its impacts, and on the early warning structure in Germany. Considering that one major aim of the paper is to compare the event with previous flooding events in Germany, it could be of advantage to briefly introduce these flooding events. This would improve the reader's understanding on the context of the previous events that may support the understanding of similarities and differences identified between the historic and the recent event. It could be included in form of a table or short paragraph in the introduction or methods chapter.

*Response 2: Thank you for this suggestion. A table will be added in the revised version.*

Results/Discussion

The section presents the extensive results of the survey by partly comparing them to previous flooding. It could be of advantage to have on section dedicated (e.g., between 3.2 and 3.3) on the major similarities/differences identified between the events and discuss these. Considering also the following question: are there aspects that actually improved during the past decades?

*Response 3: Since the flood events are so different with regard to their dynamics it is difficult to identify aspects that have continuously improved over time. We will try to add aspects that are related to the dissemination, receipt and interpretation of warning messages, since these are in the focus of our research.*

Conclusion

The conclusion could be more specific in terms of recommendations for the future (e.g., 'communication […] have to be considerably enhanced' –> you could specify/list how it should be enhanced).

*Response 4: As also suggested by referee 1 we will carefully revise the conclusion section and will particularly pay attention to more specific recommendations, e.g. for different modes of communication.*

Also, it could be interesting to see who these recommendations would be addressed to - who would be in charge of addressing the criticalities identified in this research.

*Response 5: Since different levels of governance are involved in the warning process all these levels have to be addressed – with, however, different, i.e. tailored key messages. We will add some ideas.*

I also agree that there shouldn't be any references (incl. to figures) in the conclusion.

*Response 6: See also response 10 to the other review. We still think that references to figures and tables could help the reader connect things and do not harm. According to the reviewer's preferences we will, however, reduce the number of references in the concluding section.*

Lastly, the last sentence (L324-325) is of high importance, but this topic was not discussed in the paper.

*Response: We will rephrase this sentence and propose this as a future research topic.*

*Thank you for the valuable comments.*

---

## Author Response (AR1)

**Responses to comments on egusphere-2022-244**

**"Performance of the flood warning system in Germany in July 2021 – insights from affected residents" by Annegret H. Thieken et al., EGUsphere, 2022**

In the text below we provide a point-by-point response to the comments made by the referees. For clarity, our responses are written in Italics.

**Referee 1: Martijn Kuller (Referee)**

**https://doi.org/10.5194/egusphere-2022-244-RC1**

Referee comments

This paper presents the results of a survey held among the residents of a recent severe flood event in Germany. It is a valuable piece of work both for the academic community interested in risk communication and for authorities to draw lessons from and improve practices in the future. While the overall quality of the work sufficient to warrant publication, there are a few issues that need to be addressed. Most importantly, the structure of the paper could benefit from clearer research questions, hypotheses and a methods and results section following this structure.

*Response 1: Thank you for valuing our work. In the revised version, we considered changes in the structure of the paper and rephrased our research questions clearer as outlined below. Initially, we did not follow a distinct hypothesis-based research design, but were more explorative. However, since we had some implicit hypotheses, we mention them explicitly in the revised paper. We also carefully re-considered the structure of the individual sections of the article.*

Overall the language is good and the paper reads well, but some phrases are rather "spoken" language than written language style (e.g., line 39: "save people's lives". I would rewrite to: "prevent fatalities" or "prevent loss of life". Second example in line 79: "…how the targeted population…". I would change to: "…warning perception by the target population…"). The article would benefit from review by a native English writer. Below I provide further details on this, as well as other comments by section, and then detailed comments in a table.

*Response 2: The sentences pointed out by Reviewer 1 were rephrased in the revised version. The accepted manuscript will be proofread by a native speaker before the final publication.*

Introduction

This section has a clear structure and explains the background, the need for the research and the research aim clearly. However, detailed research questions and hypotheses are missing, which are needed to justify the methods used (types of data (indicators) gathered, methods of data collection etc.).

*Response 3: In the revised version we explain our research approach and underlying hypotheses clearer. Accordingly, we extended the last paragraph of the introduction, which now reads as follows (see line 89ff):*

*"As part of a broader post-event investigation, this paper aims to analyse how the warning system in July 2021 performed – also in comparison to other flood events in Germany that are summarized in Table 1. The evaluation of the performance of the warning system is mainly*

*based on an online-survey in the affected regions and focusses on three research questions (RQ): RQ1) How many people received a warning before they were in danger? RQ2) How well did people trust and understand the warnings? RQ3) How did people respond to the warnings and how did they perceive the effectiveness of their action?*

*As indicated by Thieken et al. (2022) for the river flood of June 2013 in comparison to the pluvial/flash floods of May/June 2016, the performance of Germany's FFWRS differs per flood type. For pluvial and flash floods in 2016, there was a higher a share of affected people who were not warned, warning times were shorter and the situational knowledge was poorly developed among affected residents (Thieken et al., 2022). Given the severe impacts in 2021, we hypothesize that the performance of Germany's FFWRS in July 2021 was even worse than during recent pluvial and flash floods (see Table 1 for brief event descriptions) with regard to the dissemination of the warning messages and people's situational knowledge on protective behaviour. Since elderly people were considerably overrepresented among the flood fatalities of 2021 (Kron et al., 2022), we expect that the receipt of warnings, the situational knowledge and the perceived effectiveness of protective behaviour is influenced by the age of respondents next to the event's magnitude. The flood magnitude of July 2021 was exceptionally high as estimations of precipitation indices and of return periods of the discharge along the river Ahr revealed (Lengfeld et al., 2022; Vorogushyn et al., 2022). Therefore, we further hypothesize that damage-reducing behaviour was not perceived as effective by the respondents.*

*Following an explorative approach, we finally discuss as a fourth research question (RQ4) how to further improve the FFWRS based on the outcomes of the analyses and the views and wishes of the population affected in July 2021."*

*The results section has been structured according to the four research questions. Please note that the question on how people responded and perceived their protective action was added due to comment/response 7. The methods section was adapted/rewritten accordingly.*

Data and Methods

It is stated that the survey was distributed via facebook, and a press release was sent through the newspaper. However, it seems like the survey could only be accessed through facebook. Why did the authors choose this method, and how did they make sure this didn't lead to a biased sample, as facebook is only used by a specific demographic, leaving out other specific demographics that might be important. Beyond age, general techsavvyness and online presence play a role in the accessibility of the survey to people using facebook. If the authors somehow tried to correct for this bias, please provide some information about how this was done. If not, please discuss how the method might have influenced the outcome and what this means for the conclusions.

*Response 4: This is a misunderstanding. The online questionnaire was programmed in SoSci Survey and was accessible through a link outside of Facebook. Facebook was just used to advertise our survey. We clarified this in the revised manuscript. To avoid bias among respondents we also used traditional media like newspapers to draw attention to our survey (via a press release) and directly informed all mayors in the affected regions about our survey together with a plea to mention the survey in local newsletters. We saw a response to all advertising activities in the number of respondents to our survey. The age range presented in the paper illustrates that we managed to address elderly people, too, who are, however, underrepresented in the sample, although elderly people are increasingly online and use internet sources. We added this information in the paper.*

*We would like to highlight that we decided for an online survey in order to allow respondents a voluntary participation in the survey. This was an important ethical aspect to us given the rather short time-span after the flood event, when many people were still in the immediate recovery process.*

Also, it is explained what the questions in the survey were regarding, however I am missing some explanation as to why these questions were posed (what were the exact research questions the authors want to answer with the set of questions they selected, including the demographical questions), and what were they expecting to find? Part of this information could already be provided in the introduction. Furthermore, is the survey accessible to the reader somehow? It would be good to add them to supplementary materials for reference and future use by the readers.

*Response 5: Research questions and hypotheses are now better outlined in the introduction so that the contents of the questionnaire should be more understandable and self-evident. A translated version of the questionnaire is now provided in the Appendix. Moreover, all variables including their codings and descriptive statistics are provided in the Appendix, Table A1.*

The Likert scales used in the survey are analysed statistically, using the answers as numerical values. Although this is common practice, it is strictly speaking not valid, as these numbers are only category indicators, and not actual numerical values of the answer, and can thus not be used for statistical analysis (drawing mean, sd etc).

*Response 6: We use scales that are only verbalized at the end points as shown in Table A1. According to the (German) textbook of Porst (2014: Der Fragebogen, 4th edition, chapter 6), data from such a scale can be regarded as interval-scaled data (being ordered with equal distances between the values, but without having an absolute zero point). Therefore, it is valid to treat them as numerical data as it is common practice, as also stated by the reviewer.*

Results and discussion

If the factors investigated and presented in Table 2 explain so little in the receipt of an official warning, what then is (or could be) the most important predictor for this? All the results are self-reported data from the survey. I wonder how this self-reported data compares to behaviour and if this could be discussed in this section (e.g., how do the numbers of "knowing what to do" compare to the observed behaviour of people on the day?). Would there be any such comparison possible here in the discussion?

*Response 7: Just for clarity: Table 2 (now Table 3) shows the results of the regression on the receipt of the warning, while Table 3 (now Table 4) shows the regression on situational knowledge about protective behaviour (knowing what to do). For empirical data in social sciences, the latter has a good R² of 33 %.*

*We agree that it would be great to have some independent data to compare our results with. Unfortunately, there is only anecdotal evidence, which was now added in the manuscript on line 248ff. For example, in the most affected district of Ahrweiler in Rhineland-Palatinate around 18% of the population have registered at KATWARN, the commonly used warning app in this district. A more detailed analysis of our data for this district reveals that around 20% of the people from this district in our survey report that they had been warned by this app. In addition, the time of having received the first warning as reported in the survey also matches to the officially released warning message.*

*Furthermore, during an interview with a representative of the regional broadcast in April 2022 we discussed the percentage of people, who were not warned. That number was then compared to data on the general coverage of TV and radio. According to the media expert a higher percentage of people would only have been reached if the warnings and the upcoming event had been addressed in the programme for a few days before the event happened and by using easily interpretable stories and images. The fact that warnings of slow onset fluvial floods like the one in 2013 are much more successful (as is shown in Thieken et al. 2022 and in this paper, too) was explained by the better coverage in the media for several days and first stories of affected places and people in the upstream areas.*

*Hard facts and data are, however, absent, but the aspects mentioned above are now discussed in the revised paper (line 248ff). In general, behavioural observations during the actual event would have been neither possible nor ethical. To further address this point we therefore added new analyses of the survey data in section 3.3. First, we summarize the non-responsive proportion among respondents as well as the activities that people reported and secondly, we added a third regression showing influencing factors on the perceived effectiveness of risk-reducing behaviour. This complements the picture.*

The results presented in Figure 5 show very little variation. This is a common problem with asking respondents what they want, choosing from a list of options: they are going to want it all. Such results are not very helpful when resources are scarce and trade-offs have to be made (in this case, we want to keep a warning clear and concise). A better way to measure such preferences could therefore be asking respondents to rank or to distribute a limited number of points among the options.

*Response 8: Thank you for this remark and the suggestion which we will consider in a next survey on this topic. Still, we think that our data are of use, e.g., with regard to the expected amount of rainfall. Here a clear mismatch between the frequency of this information in the received warning messages and the wishes for the future becomes apparent. Note that this figure is now Figure 6.*

A section on further research is missing. I already made some recommendations in this review, e.g., comparing self-reported information to actual behaviour (if this is not going to be implemented by authors).

*Response 9: We added ideas on future research in the concluding section which was completely rewritten.*

Conclusions

Because the research questions weren't completely clear (see other comments), the conclusion doesn't naturally follow from the rest of the paper. When this issue is resolved, the conclusion might be written in a more concise manner, going back to these research questions.

There are many references to the main text as well as other papers in the conclusion. I recommend rewriting the conclusion according to my above instructions, and avoid any references, as a conclusion should stand on its own, not repeat the results, but rather state the significance and meaning of results. As it stands, the conclusion mostly summarises the main text.

*Response 10: Thank you for this suggestion. However, we partly disagree. A conclusion should or could start with repeating the research aim and the main findings, then proceed with implications (while accounting for the inherent uncertainties) and finally addressing future research topics. Following this structure, references to other works or to tables/figures in the paper might help the reader to connect things. According to the reviewer's preferences we have, however, reduced the number of references in the concluding section considerably.*

Furthermore, I think the paper could benefit greatly from a separate section (within conclusion or at the end of the discussion) that summarises the most important recommendations from this research. This research is very practical and applied, and has great potential to aid authorities around the world.

*Response 11: In our view these recommendations are part of the conclusions as outlined above. Consequently, we changed the title of the section in "Conclusions and Recommendations".*

Please find detailed comments in the table below:

Line 39-40: One or two sentences discussing why the performance might have been so good for these events could benefit this section here.

*Response 12: We now added some figures on the performance of FFWRS (line 44-49):*

*"Worldwide, the effectiveness of early warning systems to save lives was impressively demonstrated in the flood-prone country of Bangladesh: while a cyclone in 1999 claimed around 10,000 deaths, warning and evacuation reduced the death toll to 38 lives in 2013 (Hallegatte et al., 2020). Recent cyclones confirmed the success of the warning and response system (Ferdous et al., 2020). For Europe, Hallegatte (2012) estimated that weather information and warnings have annually saved hundreds of lives and 460 million to 2.7 billion Euros of losses, while creating even higher benefits by optimized production in weather-sensitive sectors."*

Line 153-158: This belongs more in the methods section, see previous comments.

*Response 13: These sentences were shifted to the method section that now contains details about the regression analyses.*

Line 159: Results are presented for which no methods have been described (logistic regression is mentioned). This should be added to the methods section (also see previous comments about methods).

*Response 14: We now explain the different types of regression analyses that we apply in the paper in Section 2.*

Line 160: Please explain how to interpret odds ratios.

*Response 15: We agree with the reviewer that it is useful to explain how to interpret the odds ratios reported in Table 3. Accordingly, we have added the following explanation to the text (line 162-165): "As an intuitive interpretation of regression coefficients is difficult for logistic regressions, we provide odds ratios as a measure of the effect size, which are easier to*

*interpret. An odds ratio above 1 indicates that, as the explanatory variable increases, the odds (or likelihood) of the dependent variable occurring also increases. Conversely, an odds ratio below 1 indicates that, as the explanatory variables increases, the likelihood of the dependent variable occurring decreases."*

Table 2 and 3: From the table it isn't clear how gender impacts the prediction on knowing what to do. Is it male or female that increases the knowing what to do (explain in the table).

*Response 16: The coding is explained in Table A1; we now added this information in the headings of Tables 2 (now Table 3) and 3 (now Table 4).*

Line 184: Is there data available to make a more precise comparison between the warnings issued by the different authorities and the warnings received by the population as reported through the survey? This could shed light on the effectiveness of the dissemination by the authorities. (a hint to this information is given in the text in lines 183-184).

*Response 17: Please have a look at response 7 above. We haven't done this regional analysis for the whole region, which is quite large, but we highlighted some findings for the district of Ahrweiler as example in section 3.1. A publication on regional differences in the warning communication of the 2021 event is currently in preparation.*

Line 189: The authors state that they find the trust in the credibility high. I would argue that 9% of people not believing a warning is quite a high number in an emergency situation, and less than half of the people stating they find the warning highly credible is to my perception quite low. From a government-issued warning, I would strive for credibility numbers close to 100%.

*Response 18: We agree and rephrased the sentence. There's anecdotal evidence that a lot of people distrusted the predicted magnitude, not the warning itself.*

Line 198: This sentence is confusing, please re-write.

*Response 19: The sentence now reads (line 277-281): "In addition, some respondents complained that too many warnings on Covid-19 were disseminated via the most popular warning app NINA, which was tiring and lowered their attention to warning messages. Above all, in the week prior to the severe flood event there were already warnings for heavy rain in parts of the affected region, but no serious flooding happened. False alarms are known to commonly lower trust in warnings."*

Line 210: This is more a concluding remark.

*Response 20: This aspect was shifted to the (revised) concluding section.*

Fig. 5 and 6: Meaning of the numerical scale points need to be explained in the caption. Furthermore, as indicated before, it is strictly speaking not correct analysing and presenting this data in this way.

*Response 21: Our apologies for this. The meanings of the scales are now provided in the headings. Please note that these figures are now 6 and 7. See our comment (response 6) on the validity above.*

Line 298: It is not common to refer to tables and figures in the conclusion if not strictly necessary.

*Response 22: see response 10*

*Thank you very much for the detailed comments to our paper.*

**Responses to comments on egusphere-2022-244**

**"Performance of the flood warning system in Germany in July 2021 – insights from affected residents" by Annegret H. Thieken et al., EGUsphere, 2022**

In the text below we provide a point-by-point response to the comments made by the referees. For clarity, our responses are written in Italics.

**Referee: Anonymous Referee #2**

**https://doi.org/10.5194/egusphere-2022-244-RC2**

Referee comments

This paper provides very important insights into the early warning, preparedness, and response of the German flood event in 2020. The major aim of the paper is to analyze the operation of the warning system in July 2021 and to compare survey results to survey results of historic floodings in Germany. The results presented in this paper will be of high value to improve the disaster management in Germany.

I agree with the major recommendations provided in the comment RC1; thus, below, I will provide only a few suggestions in addition to the already stated recommendations.

*Response 1: Thank you for valuing our work.*

Introduction/Methods

The introduction provides a clear and in-depth overview on the event, its impacts, and on the early warning structure in Germany. Considering that one major aim of the paper is to compare the event with previous flooding events in Germany, it could be of advantage to briefly introduce these flooding events. This would improve the reader's understanding on the context of the previous events that may support the understanding of similarities and differences identified between the historic and the recent event. It could be included in form of a table or short paragraph in the introduction or methods chapter.

*Response 2: Thank you for this suggestion. A table with brief event descriptions and available survey data has been added to the revised version (see Table 1).*

Results/Discussion

The section presents the extensive results of the survey by partly comparing them to previous flooding. It could be of advantage to have on section dedicated (e.g., between 3.2 and 3.3) on the major similarities/differences identified between the events and discuss these. Considering also the following question: are there aspects that actually improved during the past decades?

*Response 3: Since the flood events are so different with regard to their dynamics it is difficult to identify aspects that have continuously improved over time. We tried to add some aspects that are related to the dissemination, receipt and interpretation of warning messages in the section 3.1 to 3.3, since these are in the focus of our research.*

Conclusion

The conclusion could be more specific in terms of recommendations for the future (e.g., 'communication […] have to be considerably enhanced' –> you could specify/list how it should be enhanced).

*Response 4: As also suggested by referee 1 we completely rewrote the conclusion section and paid particular attention to more specific recommendations.*

Also, it could be interesting to see who these recommendations would be addressed to - who would be in charge of addressing the criticalities identified in this research.

*Response 5: Since different levels of governance are involved in the warning process all these levels have to be addressed – with, however, different, i.e., tailored key messages. We now tried to be more specific in this regard.*

I also agree that there shouldn't be any references (incl. to figures) in the conclusion.

*Response 6: See also response 10 to the other review. We still think that references to figures and tables could help the reader connect things and do not harm. According to the reviewers' preferences we, however, reduced the number of references in the concluding section considerably.*

Lastly, the last sentence (L324-325) is of high importance, but this topic was not discussed in the paper.

*Response: We rephrased this sentence as part of a proposal for future research.*

*Many thanks for your valuable comments.*

---

## Author Response (AR2)

**Responses to comments on egusphere-2022-244**

**"Performance of the flood warning system in Germany in July 2021 – insights from affected residents" by Annegret H. Thieken et al., EGUsphere, 2022**

Dear editor, dear Daniela, dear reviewers,

thank you very much for accepting our revised version of the paper. We would like to emphasize again that the comments of the two reviewers were very helpful and really led to an improved manuscript.

With regard to the technical request ("I am referring to the request of one of the referee to add English references on the use of the Likert scale, in addition to the German book of Porst."), we now added in the main text the following sentences and references. Besides the German textbook, we provide a recent internationally peer-reviewed paper on this issue (lines 187ff in the revised paper):

*"In general, data from rating scales that are end labelled are usually assumed to be equidistant, i.e., interval-scaled, according to Porst (2014), which was recently confirmed by Höhne et al. (2021) for questions on income (in-)equalities. Therefore, the survey data were mainly treated as quantitative data although in principle the equidistance of each rating scale needs validation."*

*References:*

*Höhne, J.K., Krebs, D., and Kühnel, S.-M.: Measurement Properties of Completely and End Labeled Unipolar and Bipolar Scales in Likert-type questions on Income (In)equality, Social Science Research, 97, 102544, doi.org/10.1016/j.ssresearch.2021.102544, 2021.*

*Porst, R.: Fragebogen – ein Arbeitsbuch, Springer VS, Wiesbaden, 4th edition, DOI 10.1007/978-3-658-02118-4, 2014 (in German).*

We hope that this meets your expectations and we are looking forward to the production process.

Best regards

Annegret Thieken (on behalf of all co-authors)